# Learning Some Popular Gaussian Graphical Models without Condition Number Bounds

**Jonathan Kelner**[*]
MIT

**Frederic Koehler**[†]
MIT

**Raghu Meka**[‡]
UCLA

**Ankur Moitra**[§]
MIT

## Abstract

Gaussian Graphical Models (GGMs) have wide-ranging applications in machine learning and the natural and social sciences. In most of the settings in which they are applied, the number of observed samples is much smaller than the dimension and they are assumed to be sparse. While there are a variety of algorithms (e.g. Graphical Lasso, CLIME) that provably recover the graph structure with a logarithmic number of samples, to do so they require various assumptions on the well-conditioning of the precision matrix which preclude long-range dependencies, present in many settings of interest.

Here we give the first fixed polynomial-time algorithms for learning attractive GGMs and walk-summable GGMs with a logarithmic number of samples and without any such assumptions. In particular, our algorithms can tolerate strong dependencies among the variables. Our result for structure recovery in walk-summable GGMs is derived from a more general result for efficient sparse linear regression in walk-summable models without any norm dependencies. We complement our results with experiments showing that many existing algorithms fail even in some simple settings where there are long dependency chains. Our algorithms do not.

## 1 Introduction

A Gaussian Graphical Model (GGM) in $n$ dimensions is a probability distribution with density

$$p_X(x) = \frac{1}{\sqrt{(2\pi)^n \det \Sigma}} \exp\left(-(x-\mu)^T \Sigma^{-1}(x-\mu)/2\right)$$

where $\mu$ is the mean and $\Sigma$ is the covariance matrix. In other words, it is just a multivariate Gaussian.

What makes the multivariate Gaussian so interesting as a graphical model is that its conditional independence structure is completely encoded in the *precision matrix*, $\Theta = \Sigma^{-1}$. For a given $\Theta$, if we let $G$ be the graph with vertex set $[n]$ and an edge between vertices $i, j$ iff $\Theta_{ij} \neq 0$, then $X \sim N(\mu, \Sigma)$ exhibits the following remarkable *Markov property* with respect to $G$: for *any* set $S \subset [n]$, if removing the nodes $S$ from $G$ disconnects nodes $i$ and $j$, then $X_i$ and $X_j$ are conditionally independent given $X_S$. This is a very special and useful algebraic property, because

---

[*]Department of Mathematics, Massachusetts Institute of Technology. Email: kelner@mit.edu. This work was partially supported by NSF Award CCF-1565235

[†]Department of Mathematics, Massachusetts Institute of Technology. Email: fkoehler@mit.edu. This work was supported in part by Ankur Moitra's ONR Young Investigator Award.

[‡]Department of Computer Science, UCLA. Email: raghum@cs.ucla.edu. This work was supported by NSF CAREER Award CCF-1553605.

[§]Department of Mathematics, Massachusetts Institute of Technology. Email: moitra@mit.edu. This work was supported in part by NSF CAREER Award CCF-1453261, NSF Large CCF-1565235, a David and Lucile Packard Fellowship and an ONR Young Investigator Award.

it means that if we can just estimate the *support* of the matrix $\Theta$ accurately, we can potentially discover a lot of valuable information about the random variables $X_1, \ldots, X_n$. It was for this reason that Dempster [1] initiated the study of learning GGMs in the 1970s.

GGMs have wide-ranging applications in machine learning and the natural and social sciences where they are one of the most popular ways to model statistical relationships between observed variables. For example, they are used to infer the structure of gene regulatory networks (see e.g. [2, 3, 4, 5]) and to learn functional brain connectivity networks [6, 7]. In causal inference, GGM routines are often used as a basic subroutine when attempting to learn causal models (see e.g. [8]), because it corresponds to looking for the *moral graph* of the directed model, which (1) greatly constrains the possible set of causal structures which need to be considered, and (2) can be learned from purely observational data, unlike the original causal structure. It is important to note that in most of the settings in which GGMs are applied, the number of observed samples is small compared to the dimensionality of the data. This means that in practice, it is only possible to learn the GGM in a meaningful sense under some sort of sparsity assumption. We will make the assumption that the rows and columns of $\Theta$ are $d$-sparse – i.e., the case where the dependency graph $G$ has maximum degree at most $d$.

From a theoretical standpoint, there is vast literature on learning sparse GGMs under various assumptions. Many approaches focus on *sparsistency* – where the goal is to learn the sparsity pattern of $\Theta$ assuming some sort of lower bound on the strength of non-zero interactions. This is a natural objective because once the sparsity pattern is known, estimating the entries of $\Theta$ is straightforward (e.g. one can use ordinary least squares); because of this, the problems of learning GGMs and sparse linear regression are very closely related. A popular approach to learning GGMs is the Graphical Lasso[5] [9] which solves the following convex program:

$$\max_{\Theta \succ 0} \left( \log \det(\Theta) - \langle \widehat{\Sigma}, \Theta \rangle - \lambda \|\Theta\|_1 \right)$$

where $\widehat{\Sigma}$ is the empirical covariance matrix and $\|\Theta\|_1$ is the $\ell_1$ norm of the matrix as a vector.

It is known that if $\Theta$ satisfies various conditions, which typically include an assumption similar to or stronger than the restricted eigenvalue (RE) condition (a condition which, in particular, lower bounds the smallest eigenvalue of any $2d \times 2d$ principal submatrix of $\Sigma$) then Graphical Lasso and related $\ell_1$ methods can succeed in recovering the graph structure (see e.g. [10, 11]). For the Graphical Lasso itself, under some incoherence assumptions on the precision matrix (stronger than RE), it has been shown [12] that the sparsity pattern of the precision matrix can be accurately recovered from $O((1/\alpha^2)d^2 \log(n))$ samples; here $\alpha$ is an *incoherence parameter* and we are omitting the dependence on some additional terms. We emphasize that this is only the best known theoretical guarantee — the performance in real life often seems better than this pessimistic bound.

Another popular approach to learning GGMs is the CLIME estimator which solves the following linear program:

$$\min_{\Theta} \|\Theta\|_1 \text{ s.t. } \|\widehat{\Sigma}\Theta - I\|_\infty \leq \lambda$$

The analysis of CLIME assumes a bound $M$ on the maximum $\ell_1$-norm of any row of the inverse covariance (given that the $X_i$'s are standardized to unit variance). This is also a type of condition number assumption, although with respect to a different geometry than RE: more precisely, since

$$M = \max_{\|u\|_\infty \leq 1} \|\Theta u\|_\infty$$

it can be thought of as the condition number of $\Sigma$ when viewed as an operator mapping $\ell_\infty \to \ell_\infty$; this can be smaller than the normal Euclidean condition number. CLIME succeeds at structure recovery when given

$$m \gtrsim CM^4 \log n$$

samples (here for simplicity we are assuming the entries $\Theta_{ij}$ are either zero or bounded away from zero by an absolute constant $c$ so that $M = \Omega(d)$).

While these works show that sparse GGMs can be estimated when the number of samples is logarithmic in the dimension, there is an important caveat in their guarantees. They all need to assume

that $\Theta$ is well-conditioned, and differ mainly in the strength of their assumption: roughly speaking, one of the stronger assumptions[6] used in this literature is that $\Theta$ is well-conditioned in the usual sense, and the weakest is the $\ell_\infty \to \ell_\infty$ condition number bound assumed by CLIME. This is often heuristically justified by the belief that a small condition number is information-theoretically required for structure recovery to be possible. However, and as we will discuss later, recent works have pointed out that this is actually not the case — the correct information-theoretic condition is *significantly weaker* than even the assumption which CLIME makes. Indeed, the fact that bounded condition number is not the right assumption for structure recovery is hinted at by the fact that it does not behave nicely under benign operations like rescaling individual variables. In the high-dimensional setting, bounded condition number can be a somewhat strong condition: in particular, this assumption is violated by simple and natural models (e.g. a graphical model on a path such as a time series), where these bounds turn out to be polynomial in the dimension.

In this paper, we study some fundamental classes of GGMs and show how to learn them efficiently in the low-sample regime under the correct information-theoretic assumption, even when they are ill-conditioned. We also complement our results with examples that break both previous algorithms and our own algorithms for learning general sparse GGMs. This leaves open the interesting question (raised in [14], and closely related to similar questions about sparse linear regression [15]) of whether some sparse GGMs may be computationally hard to learn with so few samples. Finally, we show experimentally that popular approaches, like the Graphical Lasso and CLIME, do in fact need a polynomial in $n$ number of samples even in some relatively benign examples (and where our algorithm does succeed).

Our work was motivated by a recent paper of Misra, Vuffray and Lokhov [14] which studied the question of how many samples are needed *information-theoretically* to learn sparse GGMs in the ill-conditioned case. They required only the following natural non-degeneracy condition (which also appeared in [16, 17]): that for every $i, j$ with $\Theta_{ij} \neq 0$, we have a lower bound on the *conditional partial correlation*[7] below:

$$\kappa \leq \frac{|\Theta_{ij}|}{\sqrt{\Theta_{ii}\Theta_{jj}}} = \frac{|\mathrm{Cov}(X_i, X_j \mid X_{\sim i,j})|}{\sqrt{\mathrm{Var}(X_i|X_{\sim i})\mathrm{Var}(X_j|X_{\sim j})}}.$$

Intuitively, this assumption means that if we have already observed all of the coordinates of $X$ except for $X_i$ and $X_j$, then the remaining randomness over $X_i$ and $X_j$ has a correlation coefficient of at least $\kappa$. This condition is the correct one because: (1) the absence of an edge in the GGM exactly corresponds to zero partial correlation in the above sense, (2) it has the correct symmetries — it is not affected by rescaling of any coordinate, and (3) it is the same condition which is needed in the classical, low-dimensional OLS regression $t$-test [18] to successfully reject the null hypothesis that the true coefficient of $X_j$ is zero when regressing $X_i$ off of $X_{\sim i}$.

Crucially, this assumption could be much weaker than any condition number bound, because it allows for the random variables to be strongly correlated. Here is a basic example (from [14]): suppose we have three Gaussians $X_1, X_2$ and $X_3$ where $X_1$ is heavily correlated with $X_2$. In this case, the condition number of $\Sigma$ will explode as $X_1$ and $X_2$ become more correlated. Nevertheless, it remains possible to test if there is a $\kappa$-nondegenerate edge between $X_1$ and $X_3$, as long as we correctly adjust for the effect of $X_2$. In contrast, if we were unaware of the value of $X_2$, it would be very difficult to test for the same edge between $X_1$ and $X_3$, because the $X_1$ and $X_2$ edge contributes a very large amount of variance to $X_1$.

The work of [14] exhibited an algorithm achieving this requirement — more precisely, they showed that it is always possible to estimate the graph structure with

$$m \geq C \frac{d}{\kappa^2} \log n$$

samples *without requiring any additional assumptions*, clarifying that further condition number assumptions are indeed unnecessary. On the other hand, the result of [17] gives an information-

theoretic lower bound[8] of $\Omega((1/\kappa^2)\log n)$ on the sample complexity for structure recovery. To summarize, the upper bound of [14] differs from the lower bound of [17] by exactly a factor of $d$ (it is unknown what the optimal dependence is) and otherwise is optimal.

However, the algorithm of [14] runs in time $n^{O(d)}$, making it impossible to run except for small instances. This is because their algorithm is based on a reduction to a sequence of sparse linear regression problems that can all be ill-conditioned. It is believed that such problems exhibit wide gaps between what is possible information theoretically and what is possible efficiently. For instance, it is known that the general *sparse linear regression* problem under fixed design is **NP**-hard[9] [19, 15]. Misra et al. solve the sparse linear regression problems using exhaustive search over $d$-size neighborhoods (hence the $n^{O(d)}$ time). This leads to the main question we study:

Can we get efficient and practical algorithms for learning GGMs (run-time $\ll n^{o(d)}$) in some natural, but still ill-conditioned, cases?

## 2 Our Results and Technical Overview

We show that for some popular and widely-used classes of GGMs—*attractive GGMs* and *walk-summable GGMs* — it is possible to achieve both logarithmic sample complexity (the truly high-dimensional setting) and computational efficiency, even when $\Theta$ is ill-conditioned.

**Attractive GGMs**

First we study the class of attractive GGMs, in which the off-diagonal entries of $\Theta$ are non-positive. In terms of the correlation structure, this means that all partial correlations are nonnegative. There are several practical motivations for studying attractive GGMs: in phylogenetic applications, observed variables are often positively dependent because of shared ancestry [20]; in various copula models that are popular in finance, we posit a latent global market variable that also leads to positive dependence [21]; see also [22] for more discussion.

A well-studied special case (which essentially captures all attractive GGMs — see Lemma 15) is the discrete Gaussian Free Field (GFF), in which case $\Theta$ is the generalized Laplacian associated to a weighted graph. This is a natural model because the Laplacian encourages "smoothness" with respect to the graph structure: see e.g. [23]; for this reason, the GFF is an important modeling tool in active and semi-supervised learning (see [24, 25, 26]); the GFF also arises in nature from a number of diverse phenomena in random walks, statistical physics, and random surfaces [27, 28, 23].

In the GFF setting, $\Theta$ will be ill-conditioned, even in the weak $\ell_\infty \to \ell_\infty$ sense, whenever some pair of vertices have large *effective resistance* between them (e.g., paths, rectangular grids, etc.,); informally, it happens when the graph has many sparse cuts.

We show experimentally (in Appendix I) that simple examples, like the union of a long path and some small cliques, do indeed foil the Graphical Lasso and other popular methods. Intuitively, this is because GFFs on a path exhibit long-range correlations that violate the assumptions used in current works — our examples show that the assumptions made in the literature are to some extent necessary for these algorithms. This analysis reveals a blind spot of the Graphical Lasso: It performs poorly in the presence of long dependency chains, which could lead to missing some important statistical relationships in applications.

We propose the following simple algorithm and show that it succeeds in learning the graph structure of attractive GGMs. This algorithm, called GREEDYANDPRUNE, does the following to learn the neighborhood of node $i$:

1. Set $S = \emptyset$ and let $\nu > 0$ be a thresholding parameter.

2. (Greedy/OMP step) Repeat the following $T$ times: set $j$ to be the the minimizer of $\widehat{\mathrm{Var}}(X_i|X_S, X_j)$ and add $j$ to $S$.

3. (Pruning step) For each $j \in S$: if $\widehat{\mathrm{Var}}(X_i|X_S) > (1-\nu)\widehat{\mathrm{Var}}(X_i|X_{S\setminus\{j\}})$, remove $j$ from $S$.

4. Return $S$ as the neighborhood of node $i$.

where $\widehat{\mathrm{Var}}$ indicates the variance is estimated from sample, using Ordinary Least Squares. A more detailed description of the algorithm is given in the Appendix. In the literature, this is called a *forward-backward method* [29].

**Theorem 1** (Informal version of Theorem 7). *Fix a $\kappa$-nondegenerate attractive GGM. The* GREEDYANDPRUNE *algorithm runs in polynomial time and returns the true neighborhood of every node $i$ with high probability with $m \geq C(d/\kappa^2)\log(1/\kappa)\log(n)$ samples, where $C$ is a universal constant.*

Our algorithm matches the sample complexity of the previous best (inefficient) algorithms for this setting [16, 14] and obtains, up to log factors, the optimal dependence on $\kappa$ for fixed $d$.

**Analysis for Attractive GGMs**  The main intuition behind the algorithm and the crux of our analysis is the following: For attractive GGMs the conditional variance of a variable $X_i$ when we condition on a set $X_S$ is a monotonically decreasing and supermodular function of $S$. This fact was previously observed in the GFF setting (independently in [26, 30]) with relatively involved proofs; we give a new, short proof of this fact using just basic linear algebra. Other works such as [31, 32] have considered supermodularity in somewhat related regression settings, but with important differences (see Further Discussion).

Given the supermodularity result, we next need to address the issue that we don't have access to actual conditional variances, but only their empirical estimates. To achieve the efficient sample complexity of Theorem 1 we carefully analyze the alignment between the true decrement of conditional variance in one step, $\mathrm{Var}(X_i|X_S) - \mathrm{Var}(X_i|X_{S\cup\{j\}})$ and the noisy empirical decrement $\widehat{\mathrm{Var}}(X_i|X_S) - \widehat{\mathrm{Var}}(X_i|X_{S\cup\{j\}})$. A subtle obstacle is that we need to control the differences $\widehat{\mathrm{Var}}(X_i|X_S) - \widehat{\mathrm{Var}}(X_i|X_{S\cup\{j\}})$ without assuming too much accuracy on the estimates $\widehat{\mathrm{Var}}(X_i|X_S)$ themselves. Fortunately, this can be shown using matrix concentration, combined with some tools from classical low-dimensional regression tests [18].

To complete the analysis, we need a new structural result for attractive GGMs which bounds the conditional variance after the first step of greedy, so that only a bounded number of iterations of greedy are required to learn a superset of the neighborhood. We prove this by reducing to the setting of discrete GFFs, where we can use an electrical argument based on effective resistances. Formally, we prove the following new structural result for walk-summable GGMs:

**Lemma 1** (Lemma 9 of the Appendix). *Suppose that $i$ is a node with $d \geq 1$ neighbors in an attractive or walk-summable GGM. Then there exists a neighbor $j$ such that*

$$\mathrm{Var}(X_i|X_j) \leq \frac{4d}{\Theta_{ii}} = 4d \cdot \mathrm{Var}(X_i|X_{\sim i}).$$

**Previous work on Learning Attractive GGMs.**  Some prior work on learning attractive GGMs have focused on the Maximum Likelihood Estimator (MLE). This was shown to exist and be unique using connections to total positivity in [33, 34], but we are not aware of any sample complexity guarantees in the context of structure learning. It also is likely broken by the same examples (see Section I) as the Graphical Lasso (since the constrained MLE is the same as the Graphical Lasso with zero regularization and a non-negativity constraint). Finally, the recent work [22] studied adaptive estimators for learning GGMs, but only for the case where the model is well-conditioned.

**Optimal Information-Theoretic Bounds.**  The previous literature leaves open the information-theoretically optimal sample complexity for learning attractive GGMs. We resolve this question: a simple estimator based on $\ell_0$-constrained least squares, which we refer to as SEARCHANDVALI- DATE, achieves sample complexity matching the information-theoretic lower bounds of [17] (whose instances can easily be made attractive) up to constants:

**Theorem 2** (Informal version of Theorem 11). *In a $\kappa$-nondegenerate attractive GGM, as long as $m = \Omega((1/\kappa^2)\log(n))$, with high probability Algorithm SEARCHANDVALIDATE returns the true neighborhood of every node $i$. This algorithm runs in time $O(n^{d+1})$.*

The results of [17] imply that $\Omega((1/\kappa^2)\log(n))$ samples are required even to distinguish the empty graph from a graph with a single $\kappa$-nondegenerate edge in an unknown location. This bound does not depend on $d$, which may appear surprising. This is possible because $d \leq 1/\kappa^2$ in $\kappa$-nondegenerate attractive GGMs — see Lemma 6. We also give a version of the above result for general models with sample complexity $O(d\log(n)/\kappa^2)$ and time complexity $O(n^{d+1})$, giving a faster alternative to [14] with the same sample complexity guarantee.

Theorem 2 is proved by a careful analysis of the signal-vs-entropy tradeoff between choosing the correct support (which is best in expectation) and an incorrect support with $k$ disagreements for each $k$. To do this we again need to study structural properties of the GGM; we establish something similar to a "margin condition" in empirical process theory [35]. Precisely analyzing the differences in empirical risk again builds upon some classical ideas in regression testing [18].

This result also identifies an important barrier to improving the information theoretic lower bound of [17], as their lower-bound instances can easily be made attractive. If this bound is not tight for general GGMs, it appears significantly new ideas will be needed to separate the sample complexity of learning attractive and non-attractive GGMs — they must rely upon the ability of negative correlations to create nontrivial cancellations.

### Walk-Summable GGMs

While attractive GGMs are natural in some contexts, in others they are not. For example, in Genome Wide Association Schemes (GWASs), genes typically have inhibitory effects too. This leads us to another popular and well-studied class of GGMs: the *walk-summable* models. These were originally introduced by Maliutov, Johnson, and Willsky [36] to explain the convergence properties of Gaussian Belief Propagation observed in practice (see also [37]).

All attractive GGMs are walk-summable, as are other important classes of GGMs like *pairwise normalizable* and *non-frustrated* models [36]. A number of equivalent definitions are known for walk-summability. The following definition is perhaps the easiest to work with: $\Theta$ is walk-summable if making all off-diagonal entries of $\Theta$ negative preserves the fact that $\Theta$ is positive definite. Perhaps less well known, walk-summable models are exactly those GGMs with Symmetric Diagonally Dominant (SDD, see Preliminaries) precision matrices under a rescaling of the coordinates — see e.g. [38, 39]. In the linear algebra literature [38], a walk-summable matrix $\Theta$ is referred to as a symmetric $H$-matrix with nonnegative diagonal.

**Analysis for learning Walk-Summable GGMs.** The analysis of learning walk-summable models is considerably different from the attractive case, because supermodularity (and even *weak supermodularity* [32]) of the conditional variance fails to hold – see Section H.1. Regardless, we are still able to prove that GREEDYANDPRUNE learns all walk-summable models with sample complexity that scales logarithmically with $n$. We also propose a variant HYBRIDMB that achieves better sample complexity.

The key idea in this analysis is that a single greedy step can serve as a kind of sparse *weak preconditioner*, roughly in terms of the $\ell_\infty \to \ell_\infty$ geometry considered in CLIME. More precisely, we show that after a single step of greedy, the unknown sparse regression vector has small $\ell_1$-norm (independent of $n$ and scaling correctly with the noise level). This is shown in the proof of Theorem 16, based on effective resistance arguments relatd to Lemma 1. The $\ell_1$-norm bound not only implies that greedy works, but also that appropriate innovations of $\ell_1$-based methods (like the Lasso) can now be guaranteed to work. We emphasize that such bounds do not hold without our "weak preconditioning" step.

Concretely, we propose an algorithm called HYBRIDMB based on this idea and show that it learns walk-summable GGMs without any condition number dependence. This algorithm does the following to learn the neighborhood of node $i$, where some technical details are left to the full algorithm description given in the Appendix:

1. (Greedy step) Set $j$ to be the minimizer of $\widehat{\mathrm{Var}}(X_i|X_j)$.

2. (Lasso with implicit weak preconditioning) Solve for $w, a$ in

$$\min_{w,a:\|w\|_1 \le \lambda} \hat{\mathbb{E}}\left[\left(X_i - \sum_{k \notin \{i,j\}} w_k \frac{X_k}{\sqrt{\widehat{\operatorname{Var}}(X_k|X_j)}} - aX_j\right)^2\right].$$

We detail the selection of $\lambda$ in the full version of the algorithm — see the Appendix.

3. (Pruning step) We perform a pruning step similar to GREEDYANDPRUNE to zero out some of the entries of $w$, and to test if $j$ is an actual neighbor.

4. Return $j$ (if it passed the test) and the remaining support of $w$ as the neighborhood of $i$.

The analysis of HYBRIDMB uses the aforementioned structural results for walk-summable models and a statistical analysis for the regression problem arising after the greedy step. The regression analysis is similar in spirit to the usual generalization bounds for $\ell_1$-constrained regression but slightly more subtle. The key insight is that the output of the algorithm is the same if we replace $X_k$ by $X_k - \mathbb{E}[X_k|X_j]$; this change of basis is unknown to the algorithm, but the analysis is much easier because $X_j$ becomes independent of the other regressors.

**Theorem 3** (Informal version of Theorem 17). *Fix a walk-summable, $\kappa$-nondegenerate GGM. Algorithm* HYBRIDMB *runs in polynomial time and returns the true neighborhood of every node $i$ with high probability given $m \ge C(d/\kappa^4)\log(n)$ samples, where $C$ is a universal constant.*

We can also prove a similar (but slightly weaker) guarantee for Algorithm GREEDYANDPRUNE — see Theorem 18. For context, we note that prior to our work, Anandkumar, Tan, Huang and Willsky [16] gave an inefficient $n^{O(d)}$ time algorithm for learning walk-summable models with similar guarantees and requiring some additional assumptions.

The above structure learning result requires $\kappa$-nondegeneracy and sparsity of the entire model. However, it is proved using the following general result for sparse linear regression, which requires only a joint walk-summability assumption:

**Theorem 4** (Informal version of Theorem 16). *Suppose that $Y = w \cdot X + \xi$ where $w$ is $d$-sparse, $\xi \sim N(0, \sigma^2)$ is independent of multivariate Gaussian r.v. $X \sim N(0, \Sigma)$, and suppose that the joint distribution of $(X_1, \ldots, X_n, Y)$ is a walk-summable GGM. Given $m$ samples from this model,* WS-REGRESSION *runs in polynomial time and returns $\hat{w}$ such that*

$$\mathbb{E}[(w \cdot X - \hat{w} \cdot X)^2] = O(\sigma^2 \sqrt{d\log(n)/m})$$

*with high probability.*

Although this result gives a "slow rate" of $\sqrt{1/m}$, it is quite different from the usual slow rate guarantee for the Lasso. The latter guarantees an upper bound on the prediction error of the form $O(\sigma RW \sqrt{\log(n)/m} + RW\log(n)/m)$ where $R$ is an $\ell_1$ norm bound on $w$ and $W$ is an $\ell_\infty$ bound on $X$, see e.g. [40, 41]. To interpret this, we can rescale the problem so that $R, W = \Theta(1)$. Then Theorem 16 guarantees error on the order of the noise level $\sigma^2$ using $O(d\log(n))$ samples – in comparison, the standard slow rate result only guarantees error on the order of $\sigma$ plus an additional term. This difference is the key to achieving structure recovery from $O(\log n)$ samples: $\sigma$ can be orders of magnitude smaller compared to $RW$ in our applications. Compared to $\ell_0$-constrained least squares, which requires runtime $O(n^d)$, the above result is computationally efficient and still has the correct dependence on $d$ and $\sigma^2$.

**General Models.** There do exist some well-conditioned GGMs which are not walk-summable. However, our analysis actually shows that our methods (GREEDYANDPRUNE, HYBRIDMB) also recover similar sample complexity bounds to [42] under their assumptions (the aforementioned $\ell_\infty \to \ell_\infty$ condition number bound) — see Theorem 19. Therefore, our results are a strict extension of the situation considered in prior work.

**Non-Gaussian Models.** It's well-known that many results for Gaussian Graphical Models can be generalized to other distributions in the following sense: if we can learn a GGM with precision matrix $\Theta = \Sigma^{-1}$, then the result will generally extend to estimating $\Theta = \Sigma^{-1}$ for $X$ with sufficiently strong concentration assumptions. The reason is that for any result which depends only on the first

two moments of $X$ (i.e. any quantity definable in terms of $\Sigma, \mu$), we can generalize it to such an $X$ by considering the Gaussian with matching first and second moments, and higher moments are generally needed only for concentration purposes.

We briefly note that the guarantees for our algorithms will also extend in this sense if, for any $w \in \mathbb{R}^n$, the sub-Gaussian constant of $w \cdot X$ is upper bounded by $C\mathrm{Var}(w \cdot X)$ for a fixed constant $C$, as the needed concentration estimates generalize [43]. On the other hand, for non-Gaussian distributions the connection between $\Theta$ and conditional independence will not generally hold.

## 2.1 Further Discussion

**GGMs vs Ising Models.** There exist parallels but also surprisingly significant differences between learning GGMs and Ising models. For Ising models, Bresler [31] gave a greedy algorithm that builds a superset of the neighborhood around each node and then prunes to learn the true graph structure using $O(f(d)\log n)$ samples and under some relatively mild assumptions. This greedy algorithm is able to perform structure learning in Ising models even when they exhibit long range correlations, which was previously considered a difficult case to analyze. However in our setting, and unlike the previously described situation for Ising models, variables have real values and can have arbitrarily small or large variance. It turns out this changes the problem dramatically, as it means that the inter-node fluctuations in the random field (which contribute to $\mathrm{Var}(X_i)$) may be orders of magnitude larger than the per-node fluctuations (corresponding to $\mathrm{Var}(X_i|X_{\sim i})$). This is exactly the setting $\sigma \ll RW$ discussed in the context of sparse linear regression.

As a result of this difference, greedy methods fail to learn general GGMs from $O(\mathrm{polylog}(n))$ samples (see Appendix J), so any analysis of greedy methods must rely on structural results for a subclass of models. The same issue comes up when learning the model directly from $\ell_1$-constrained regression guarantees as in [44, 45] — in fact, we will see in Section I that natural methods based only on $\ell_1$ regularization fail even in some relatively simple attractive GGMs (where greedy works).

**Sparse Linear Regression and Submodularity.** As previously mentioned, Das and Kempe [32] studied the problem of sparse regression without assuming the restricted eigenvalue condition. While in sparse regression, in order to learn the parameters accurately (in additive error) some bound on the condition number is needed, they studied the problem of selecting a subset of columns that maximizes squared multiple correlation (a.k.a. minimizes mean squared error). They then gave approximation guarantees for greedy algorithms under an approximate submodularity condition and assuming access to the true joint covariance matrix (in other words, they studied this as a purely algorithmic problem while ignoring sample complexity).

Our algorithm for attractive models follows the same supermodularity-based strategy, but has no knowledge of the true model except for the samples it sees. Therefore it requires a careful analysis of the interaction between the greedy iteration and noise. In the more general setting of walk-summable GGMs, we show the conditional variance does not satisfy an approximate supermodularity condition with any constant submodularity ratio. (See Remark 7.)

**Some other Related Work on Sparse Linear Regression.** In the literature on sparse regression, it is well known that the analyses of the Lasso which work well in a compresssed sensing style setting (i.e. with restricted eigenvalues, incoherent columns, etc.) is not always the correct tool to use when the coordinates of $X$ (columns of the design matrix) are highly correlated — see e.g. [46, 47, 48]. For example, the work of Koltchinskii and Minsker [49] discusses this issue in the context of Brownian motion and other situations and develops general new guarantees for $\ell_1$-penalized regression which apply under correlated design (as well as infinite dimensional settings). They consider the case where the response is a linear combination of well-separated measurements in time, which is incomparable to the situation we analyze. It would be interesting to see if the ideas used in Algorithms HYBRIDMB and GREEDYANDPRUNE can be used in some of these other settings.

## 3 Simulations and Experiments

We ran several simulations on synthetic data (i.e. generated by a true GGM), and also a small Riboflavin dataset from [50], to compare our method to those previously proposed in the literature, including the Graphical Lasso, CLIME, ACLIME and the Lasso-based Meinhausen-Buhlmann

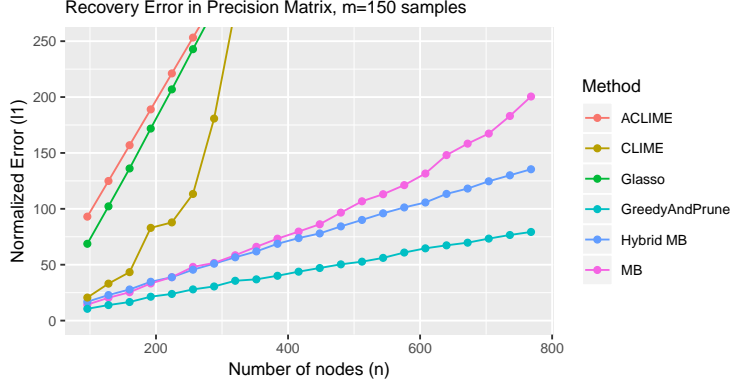

Figure 1: Normalized error (measured by $\|\hat{\Theta} - \Theta\|_1/n$), for a GGM supported on a path and some disjoint cliques Full details are given in the Appendix, where this figure is reproduced as Figure 2.

estimator. We found that both GREEDYANDPRUNE and HYBRIDMB always performed around as well as the previous methods, and greatly dominated the previous algorithms in some of the synthetic experiments.

Due to space reasons, we only include Figure 1 which shows the result of a synthetic experiment on a simple ill-conditioned attractive GGM, where the goal is to estimate the precision matrix entrywise accurately. A detailed description of all simulations and experiments performed, hyperparameter selection, etc. is given in Appendix I.

## 4 Some Difficult Examples

A natural question, given our results, is whether the GREEDYANDPRUNE algorithm could possibly learn all sparse $\kappa$-nondegenerate GGMs with $O(\log n)$ sample complexity (without requiring walk-summability). Here we answer this question in the negative. Note by the analysis from Section G.2 that if our GREEDYANDPRUNE fails to succeed with $O(\log n)$ samples, then any analysis based on bounded $\ell_1$-norm must also fail, since greedy methods always succeed if the $\ell_1$-norm is small.

It is not too hard to find examples which break the greedy method when run once from a single node, with the goal of recovering just that node's neighborhood. For example, if we take $n$ pairs of near-duplicate variables $(X_i, X_i')$ with $\mathrm{Var}(X_i) = \Theta(n)$ and $\mathrm{Var}(X_i - X_i') = \Theta(1)$ and define $Y = X_i - X_i'$ for some $i$, then using OMP to find a predictor of $Y$ will fail to find the edge from $X_i$ to $Y$ with $O(\log n)$ samples. However, if we run a greedy method to find a predictor of $X_i$, then we actually will discover this edge. In the following example, we see there are edges which are not discovered from either direction:

**Example 1** (Example breaking GREEDYANDPRUNE). *Fix $d > 2$ and let $Z_1, \ldots, Z_d$ be the result of taking $d$ i.i.d. Gaussians and conditioning on $\sum_i Z_i = 0$. Define $X_i = Z_i + \delta W_i$ and $Y_i = Z_i + \delta W_i'$ where $W_i, W_i' \sim N(0, 1)$ independently. Let $\Sigma_0$ be the covariance matrix of $X_1, \ldots, X_d, Y_1, \ldots, Y_d$ (so the $Z$ are treated as latent variables).*

*It can be checked that the GGM with covariance matrix $\Sigma_0$ remains $\kappa$ nondegenerate for a fixed $\kappa$ even as $\delta$ is taken arbitrarily small. Now consider the GGM which is block diagonal with first block $\Sigma_0$ and the second block the identity matrix, and suppose $n$ is large. If we try to learn the neighbors of $X_i$, greedy will with high probability fail to find a superset of the correct neighborhood of node $X_i$, because after conditioning on $Y_i$, the angles between the residual of $X_i$ and all of the other random variables are all near 90 degrees (going to 90 as $\delta \to 0$).*

To summarize, this example is sparse and $\kappa = \Theta(1)$ nondegenerate but GREEDYANDPRUNE fails to learn this GGM from $O(\log n)$ samples. In Appendix J give an even harder example, which experimentally seems to break all polynomial time GGM learning algorithms proposed in the literature. The following important problem, first posed in [14], remains open: are $\kappa$-nondegenerate GGMs learnable from $O(\log n)$ samples with polynomial time algorithms?

## Broader Impact

We expect our work will be most useful to theorists and practitioners interested in learning graphical models from data and related problems; for example, we hope it will raise awareness that the output of GGM learning algorithms commonly used in practice can be misleading when the data is poorly conditioned. As we established in our paper, our methods provably succeed for a subset of ill-conditioned GGMs but there remain other situations where even these methods (as well as all other popular approaches) fail. As with other linear models and machine learning tools, the algorithm can fit biases in the data. For this reason it remains necessary for practitioners to apply due diligence when interpreting the results of this, or any other, GGM learning algorithm.

## Footnotes

[5]We note that [9] did not introduce this objective (see discussion there), but rather an *optimization procedure* used to maximize it, and Graphical Lasso technically refers to this specific optimization procedure.

[6]Indeed, there are even stronger assumptions such as quantitative versions of *faithfulness* which we do not discuss but are needed to prove the correctness of the popular PC algorithm [13].

[7]Here $X_{\sim i}$ (resp. $X_{\sim i,j}$) denotes the random vector formed by deleting coordinate $i$ (resp. $i, j$) of $X$; please see Preliminaries for further details and formal definitions of conditional variances, covariances.

[8]A subtle point arises when interpreting this bound, because $d$ and $\kappa$ are closely related quantities (see e.g. Lemma 6 below). In the lower bound constructions of [17] they have $d = O(1/\kappa)$ and the term dominating their bound depends only on $\kappa$.

[9]For proper learning, where the algorithm is required to output a $d$-sparse estimator.

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
