[Supplementary Material · supp.pdf]

# A    Outline of the Appendix

Here we briefly outline the structure of the Appendix, which contains the proofs of the main Theorems as well as some simple simulations and experiments validating the theory. Each item below corresponds to a single section in the Appendix.

1. Preliminaries: we explain some fundamental facts about GGMs and fix the notation we use throughout the rest of the paper.

2. Structural results for walk-summable models: In this section, we use the connection between walk-summability, SDD matrices, and electrical circuits to establish a number of new structural results about walk-summable GGMs that will be useful for learning them. As mentioned earlier, the fundamental fact we establish in this section which is needed in all of our algorithms is that a single step of a greedy method (Orthogonal Matching Pursuit) can serve as a "weak preconditioner" for sparse linear regression, in terms of $\ell_1/\ell_\infty$ geometry. In particular, we establish the key Lemma 1 stated above.

3. Estimating changes in conditional variance: In this section, we recall the various facts we will need about ordinary least squares regression and prove a useful quantitative estimate for estimating changes in conditional variance.

4. Learning all attractive GGMs efficiently: we use further structural results about supermodularity in attractive GGMs and the results developed in the previous two sections to prove Theorem 1.

5. Information-theoretic optimal learning of attractive GGMs: In this section, we show how the result of the previous section can be improved as far as sample complexity if we are willing to sacrifice runtime, by giving a very precise analysis of a natural algorithm using $\ell_0$-constrained squares, proving Theorem 2.

6. Hybrid $\ell_1$-regression guarantees: In this section, in preparation for proving our results about learning general walk-summable models, we develop the needed statistical guarantees for a variant of the LASSO where a single coordinate in the regression is left unregularized and also give an analysis of Orthogonal Matching Pursuit in essentially the same setup.

7. Regression and structure learning in walk-summable models: In this section, we first show that supermodularity fails in walk-summable models, even if we ask for supermodularity to only hold approximately. We then proceed to establish Theorem 4 for sparse linear regression in general walk-summable models and use this result to derive Theorem 3 for structure recovery in $\kappa$-nondegenerate walk-summable models.

8. Simulations and Experiments: In this section, we compare the methods proposed in this paper to those in a number of previous works on both simulated and real data. The simulations show that all previous methods indeed fail to achieve competitive sample complexity in simple settings where the precision matrix is not well-conditioned.

9. Some difficult examples: In this short final section, we give some examples which are not walk-summable and which break both the algorithms proposed previous to this paper and in this paper as well. We show that these examples are, however, not computationally hard to learn.

# B    Preliminaries

In this section we set out some notation and basic facts about GGMs which will be used throughout.

**Notations.**    Given a GGM with precision matrix $\Theta$, $d$ will always denote the maximum degree of the underlying graph. Thus, $\Theta$ has at most $d + 1$ nonzero entries in each row. For a vector $x$ and index $i$, $X_{\sim i} = ((X_j) : j \neq i)$. For a square matix $S \in \mathbb{R}^{k \times k}$ and $I \subseteq [k]$, $S_I$ denotes the $I \times I$ principal submatrix of $S$. We will say a symmetric matrix $M$ is SDD (Symmetric Diagonally Dominant) if its diagonal is nonnegative and for every row $i$, $M_{ii} \geq \sum_{j \neq i} |M_{ij}|$. We often use the notation $\hat{\mathbb{E}}$ to denote the *empirical expectation*, i.e. expectation taken over the sample of data given to the algorithm.

We recall that conditioning on $X_i = x_i$ for any $x_i$ yields a new GGM with the precision matrix having row $i$ and column $i$ deleted. In particular, the conditional precision matrix does not depend on the value of $x_i$ chosen. Similarly, the value of the mean $\mu$ does not affect the covariance structure at all — so $\mu$ does not play an interesting role in the structure learning problem and the reader may safely assume $\mu = 0$. We summarize the facts that we use the most below.

**Fact 1** ([51]). *Let $X$ be drawn from a mean $0$ GGM with precision matrix $\Theta$. Then, for any $i$, $X_i | X_{\sim i} = x_{\sim i}$ is distributed as $N(\langle w^{(i)}, x_{\sim i}\rangle, 1/\Theta_{ii})$ where $w^{(i)}$ is the vector with $w_j^{(i)} = -\Theta_{ij}/\Theta_{ii}$.*

Thus, if we fix an index $i$, then samples $X$ from the GGM can be interpreted as a linear regression problem as $(X_{\sim i}, X_i)$ where $X_i = \langle w^{(i)}, X_i\rangle + N(0, 1/\Theta_{ii})$. This establishes the basic connection between learning GGMs and linear regression: if we can solve the above regression problem well, perhaps we can recover the non-zero entries of $\Theta$ from the coefficients. But as is well known in the literature, just fitting the coefficients using ordinary least squares is not sufficient (or necessarily possible) as we have very few samples.

By positive definiteness, we have $\Theta_{i,i} \geq 0$ and $\Theta_{i,i}\Theta_{j,j} - \Theta_{i,j}^2 \geq 0$, or equivalently $0 \leq \frac{|\Theta_{i,j}|}{\sqrt{\Theta_{i,i}\Theta_{j,j}}} \leq 1$. To identify the graph we need the present edges to not be too weak. So it makes sense to assume (following the notation of [16, 14]) there is a $\kappa > 0$ such that

$$\kappa \leq \frac{|\Theta_{i,j}|}{\sqrt{\Theta_{i,i}\Theta_{j,j}}} \leq 1 \tag{1}$$

**Definition 1** ([16, 14]). *We say a GGM is $\kappa$-nondegenerate if it satisfies (1) for all $i, j$ such that $\Theta_{ij} \neq 0$.*

**Conditional Variance.** Conditional variances of the form $\mathrm{Var}(X_i | X_S)$ play a central role in all our algorithms. We first review the basic definition and some of their properties.

**Definition 2** (Conditional Variance). *For $X$ an arbitrary real-valued random variable and $Y$ an arbitrary random variable or collection of random variables on the same probability space, let[10]*

$$\mathrm{Var}(X|Y) := \mathbb{E}[(X - \mathbb{E}[X|Y])^2].$$

By the Pythagorean Theorem, conditional variance obeys the *law of total variance* [52]:

$$\mathrm{Var}(X) = \mathrm{Var}(X|Y) + \mathrm{Var}(\mathbb{E}[X|Y]).$$

and more generally, $\mathrm{Var}(X|Y) = \mathrm{Var}(X|Y, Z) + \mathrm{Var}(\mathbb{E}[X|Y, Z]|Y)$. The last identity is also sometimes referred to as the law of total conditional variance.

The $\kappa$-nondegeneracy assumption implies a quantitative lower bound on conditional variances $\mathrm{Var}(X_i | X_S)$ when the conditioning set does not include all of $i$'s neighbors.

**Lemma 2.** *Fix a node $i$ in a $\kappa$-nondegenerate GGM, and let $S$ be set of nodes not containing all neighbors of $i$. Then*

$$\mathrm{Var}(X_i | X_S) \geq \frac{1 + \kappa^2}{\Theta_{ii}}$$

*Proof.* Let $j \notin S$ be a neighbor of $i$. By the law of total conditional variance, we have

$$\mathrm{Var}(X_i | X_S) = \mathrm{Var}(X_i | X_{\sim i}) + \mathrm{Var}(\mathbb{E}[X_i | X_{\sim i}] | X_S) = \frac{1}{\Theta_{ii}} + \mathrm{Var}(\mathbb{E}[X_i | X_{\sim i}] | X_S),$$

where in the last equality we used Fact 1. Thus, as $\mathbb{E}[f^2] \geq \mathrm{Var}(f)$, and the definition of $\kappa$-nondegeneracy

$$\mathrm{Var}(X_i | X_S) - \frac{1}{\Theta_{ii}} = \mathrm{Var}(\mathbb{E}[X_i | X_{\sim i}] | X_S) = \mathbb{E}[(\mathbb{E}[X_i | X_{\sim i}] - \mathbb{E}[X_i | X_S])^2]$$

$$\geq \mathrm{Var}(\mathbb{E}[X_i|X_{\sim i}] - \mathbb{E}[X_i|X_S]|X_{\sim j}) = \frac{\Theta_{ij}^2}{\Theta_{ii}^2 \Theta_{jj}} \geq \frac{\kappa^2}{\Theta_{ii}}$$

where the last equality follows from Fact 1 and the last inequality is by the definition of $\kappa$. The Lemma follows by rearranging. $\square$

The following basic fact about Gaussians will be useful:

**Lemma 3.** *If $X$ and $Y$ are jointly Gaussian random variables then $\mathbb{E}[X|Y] = \mathbb{E}[X] + \frac{\mathrm{Cov}(X,Y)}{\mathrm{Var}(Y)}(Y - \mathbb{E}[Y])$ and $\mathrm{Var}(X) - \mathrm{Var}(X|Y) = \frac{\mathrm{Cov}(X,Y)^2}{\mathrm{Var}(Y)}$.*

*Proof.* Because the random variables are jointly Gaussian, we know that $\mathbb{E}[X|Y]$ must be an affine function of $Y$. From $\mathbb{E}[\mathbb{E}[X|Y]] = \mathbb{E}[X]$ and $\mathrm{Cov}(\mathbb{E}[X|Y], Y) = \mathrm{Cov}(X, Y)$ the coefficients are determined, proving the first formula. Then the second formula follows from the law of total variance, $\mathrm{Var}(X) - \mathrm{Var}(X|Y) = \mathrm{Var}(\mathbb{E}[X|Y])$. $\square$

We will also use the following concentration inequality often. Recall that a $\chi^2$-random variable with $D$ degrees of freedom is just $\sum_{i=1}^{D} Z_i^2$ where $Z_i \sim N(0,1)$ are independent standard Gaussians.

**Lemma 4** (Lemma 1, [53]). *Suppose $U$ is $\chi^2$-distributed with $D$ degrees of freedom. Then $\Pr(U - D \geq 2\sqrt{D \log(1/\delta)} + 2\log(1/\delta)) \leq \delta$ and $\Pr(D - U \geq 2\sqrt{D \log(1/\delta)}) \leq \delta$. In particular, $U \leq 2D$ with probability at least $1 - \delta$ as long as $D \geq 8\log(1/\delta)$.*

## C  Structural results for walk-summable models

### C.1  Background: Walk-Summable Models are SDD after rescaling

**Definition 3** ([36]). *A Gaussian Graphical Model with invertible precision matrix $\Theta \succ 0$ is walk-summable if $D - \overline{A} \succ 0$ where $\Theta = D - A$ decomposes $\Theta$ into diagonal and off-diagonal components, and $\overline{A}$ is the matrix with $\overline{A}_{ij} = |A_{ij}|$.*

It is well-known (and immediate) that the class of walk-summable matrices includes the class of SDD matrices. Indeed, the motivation for introducing walk-summable matrices was to generalize the notion of SDD matrices.

**Definition 4.** *A matrix $M$ is symmetric diagonally dominant (SDD) if it is symmetric and $M_{ii} \geq \sum_{j:j\neq i} |M_{ij}|$ for every $i$.*

Perhaps less well-known, a natural converse holds: all walk-summable matrices are simply rescaled SDD matrices, where the rescaling is in the natural sense for a bilinear form. Furthermore, this rescaling is easy to find algorithmically (if we have access to $\Theta$), requiring just a top eigenvector computation. This result can be found explicitly in [39]; it also appears in [38] and closely related results for $M$-matrices appear in [54].

**Theorem 5** (Theorem 4.2 of [39]). *Suppose $\Theta$ is walk-summable. Then there exists a diagonal matrix $D$ with positive entries such that $D\Theta D$ is an SDD matrix.*

*Proof.* We include the proof for completeness — it is the same as in [39].

First, we observe that we can reduce to the case $\mathrm{diag}(\Theta) = \vec{1}$ by replacing $\Theta$ by $D_1 \Theta D_1$ where $D_1$ is the diagonal matrix with $(D_1)_{ii} = 1/\sqrt{\Theta_{ii}}$. Next, let $\overline{\Theta} = I - \overline{A}$ and note that when we write the decomposition $0 \prec \overline{\Theta} = I - \overline{A}$ that $\overline{A}$ has all nonnegative entries, so we can apply the Perron-Frobenius Theorem to find an eigenvector $v$ with positive entries and eigenvalue $\lambda = \|\overline{A}\| < 1$. Now define $D_2 = \mathrm{diag}(v)$, and we claim that $D_2 \Theta D_2$ is an SDD matrix. It suffices to check that $0 \leq D_2 \overline{\Theta} D_2 \vec{1} = D_2 \overline{\Theta} v$ entry-wise, and because $D_2$ is diagonal with nonnegative entries it suffices to check that $\overline{\Theta} v \geq 0$. This follows as

$$\overline{\Theta} v = (I - \overline{A})v = (1 - \lambda)v \geq 0$$

entrywise. $\square$

**Example 2.** *In Example 1 of [36] it was observed that the matrix*

$$\begin{bmatrix} 1 & -r & r & r \\ -r & 1 & r & 0 \\ r & r & 1 & r \\ r & 0 & r & 1 \end{bmatrix}$$

*itself stops being SDD when $r > 1/3$, but remains walk-summable until a little past $r = 0.39$. When $r = 0.39$, the corresponding Perron-Frobenius eigenvector for $\overline{A}$ is roughly $(0.557, 0.435, 0.557, 0.435)$ and applying the rescaling from Theorem 5 we get*

$$\begin{bmatrix} 0.310634 & -0.0945889 & 0.121147 & 0.0945889 \\ -0.0945889 & 0.189366 & 0.0945889 & 0. \\ 0.121147 & 0.0945889 & 0.310634 & 0.0945889 \\ 0.0945889 & 0. & 0.0945889 & 0.189366 \end{bmatrix}$$

*which is an SDD matrix.*

The SDD rescaling given by Theorem 5 will play a key role in our analysis. Conceptually, converting a walk-summable matrix to its SDD form is a way to take the extra degrees of freedom in the model specification (arbitrariness in the scaling of the $X_i$) and fix them in a way that is useful in the analysis – i.e. a gauge fixing. In particular, under the SDD rescaling there are meaningful relations between the different rows of $\Theta$ which fail to hold in general.

## C.2 Background: SDD systems, Laplacians, and electrical flows

**Definition 5.** *A matrix $L$ is a* generalized Laplacian *if it is SDD and for every $i \neq j$, $L_{ij} \leq 0$. We think of this graph theoretically as the Laplacian of the weighted graph with edge weights $-L_{ij}$ between distinct $i$ and $j$ and self loops of weight $L_{ii} - \sum_{j \neq i} |L_{ij}|$ at vertex $i$.*

We review the standard reduction between solving SDD systems and Laplacian systems. Suppose $\Theta$ is an SDD matrix. Then we can write $\Theta = L - P$ where $L$ is a (generalized) Laplacian having positive entries on the diagonal and nonnegative entries off the diagonal, and $P$ has negative off-diagonal entries and corresponds to the positive off-diagonal entries of $\Theta$. Now we observe that

$$\begin{bmatrix} L & P \\ P & L \end{bmatrix} \begin{bmatrix} x \\ -x \end{bmatrix} = \begin{bmatrix} \Theta x \\ -\Theta x \end{bmatrix} \tag{2}$$

and the left matrix is itself a (generalized) Laplacian matrix on a weighted graph which we will refer to as the "lifted graph".

The inverse of a Laplacian has a natural interpretation in terms of electrical flows, where the edge weights are interpreted as conductances of resistors. In this case the self loops can be thought of as resistors connected directly to electrial ground. In the next Lemma we summarize the relevant facts about this interpretation, as can be found in e.g. [55]

**Lemma 5.** *Suppose that $L$ is a (generalized) Laplacian matrix. Then if $L^+$ is the pseudo-inverse of $L$, and we define the* effective resistance $R_{\text{eff}}(i, j) := (e_i - e_j)^T L^+ (e_i - e_j)$ *then $R_{\text{eff}}$ satisfies:*

- *(Nonnegativity) $R_{\text{eff}}(i, j) \geq 0$.*

- *(Monotonicity) $R_{\text{eff}}(i, j) \leq \frac{1}{|L_{ij}|}$, and more generally $R_{\text{eff}}$ decreases when adding edges to the original adjacency matrix.*

- *(Triangle inequality) $R_{\text{eff}}(i, k) \leq R_{\text{eff}}(i, j) + R_{\text{eff}}(j, k)$ for any $i, j, k$.*

In the generalized Laplacian case, we can think of $\text{Var}(X_i | X_S)$ as being the effective resistance from node $i$ to ground when all of the nodes in $S$ are connected by wires (without resistance) to ground.

## C.3 Key structural results for Walk-Summable GGM

First we prove a fundamental fact about $\kappa$-nondegeneracy in walk-summable models, mentioned earlier: the maximum degree $d$ always satisfies $d = O(1/\kappa^2)$ in $\kappa$-nondegenerate walk-summable models. This result is tight for star graphs.

**Lemma 6.** *In a $\kappa$-nondegenerate walk-summable GGM, the maximum degree of any node is at most $1/\kappa^2$.*

*Proof.* Rescale the coordinates so that the diagonal of $\Theta$ is all-1s, and reorder them so that $X_1$ corresponds to the node of maximum degree $d$ with neighbors $2, \ldots, d + 1$. Define $\overline{\Theta}$ to be the sign-flipped version of $\Theta$ such that all off-diagonal entries are negative; by the definition of walk-summability we know $\overline{\Theta}$ is still PSD. Let $v = (1, \kappa, \ldots, \kappa) \in \mathbb{R}^{d+1}$ and $S = \{1, \ldots, d + 1\}$; then using that the off-diagonals are negative, $\kappa$-nondegeneracy we find that $\Theta_{d+1,d+1} v \leq (1 - d\kappa^2, 0, \ldots, 0)$ coordinate-wise, hence using $\overline{\Theta} \succeq 0$ we find

$$0 \leq v^T \Theta_{d+1,d+1} v \leq v^T (1 - d\kappa^2, 0, \ldots, 0) = 1 - d\kappa^2.$$

Rearranging we see that $d \leq 1/\kappa^2$. $\qquad\square$

In the remainder of this subsection we prove some key structural results about walk-summable/SDD GGM using the SDD to Laplacian reduction and the electrical interpretation of the inverse Laplacian; these results will be crucial for analyzing the algorithms for both attractive and general walk-summable GGMs.

The following key Lemma, which shows that the variance between two adjacent random variables in the SDD GFF cannot differ by too much, will be crucial in the analysis of our algorithm in non-attractive models. Why is this useful? Informally, this is because for the greedy method to significantly reduce the variance of node $i$, at least one neighbor of $i$ needs to provide a good "signal-to-noise ratio" for estimating $X_i$, and under the SDD scaling, this inequality shows that the neighbors do not have too much extra noise (compared to $|\Theta_{ij}|$ which roughly corresponds to the level of signal between nodes $i$ and $j$).

**Lemma 7.** *Suppose that $\Theta$ is an invertible SDD matrix. Let $\Sigma = \Theta^{-1}$. If $\Theta_{ij} \neq 0$, then*

$$\Sigma_{ii} \leq 1/|\Theta_{ij}| + \Sigma_{jj}.$$

*Proof.* Let $M$ be the generalized Laplacian matrix resulting from applying the SDD to Laplacian reduction from $\Sigma$, i.e. $M$ is the left hand-side of (2). Let the standard basis for $\mathbb{R}^{2n}$ be denoted $e_1, \ldots, e_n, e'_1, \ldots, e'_n$. Observe from (2) that

$$\Sigma_{ii} = e_i^T \Theta^{-1} e_i = e_i^T M^+ (e_i - e'_i) = \frac{1}{2}(e_i - e'_i)^T M^+ (e_i - e'_i).$$

Let node label $i$ be the node corresponding to $e_i$ in the graph corresponding to $M$, and label $i'$ be that corresponding to $e'_i$. Observe that in the graph corresponding to $M$, either $i$ is adjacent to $j$ and $i'$ is adjacent to $j'$, or $i$ is adjacent to $j'$ and $i'$ is adjacent to $j$. Let $r = R_{\text{eff}}(i, j)$ in the first case and $r = R_{\text{eff}}(i, j')$ in the second case. By the triangle inequality (Lemma 5) and monotonicity of effective resistance (Lemma 5),

$$2\Sigma_{ii} = R_{\text{eff}}(i, i') \leq 2r + R_{\text{eff}}(j, j') \leq 2/|\Theta_{ij}| + 2\Sigma_{jj}$$

which proves the result. $\qquad\square$

**Remark 1.** *Note that the above Lemma is for $\Theta$ under the true SDD scaling. It would not make sense for general $\Theta$, because the left hand and right hand sides do not behave the same way when we rescale $X_i$ and $X_j$.*

The following two lemmas show that in a SDD GGM, the variance of a single node can be bounded as long as we condition on any of its neighbors. In comparison, if we don't condition on anything then the variance can be arbitrarily large: consider the Laplacian of any graph plus a small multiple of the identity.

**Lemma 8.** *Suppose that $i$ is a non-isolated node in an SDD GGM. Then for any neighbor $j$ it holds that*

$$\text{Var}(X_i | X_j) \leq \frac{1}{|\Theta_{ij}|}$$

*Proof.* This result can be obtained from the previous Lemma 7 by taking an appropriate limit which sends $\Sigma_{jj} \to 0$. We give an alternate and direct proof below.

Apply the SDD to Laplacian reduction to the precision matrix (with row and column $j$ eliminated) as in Lemma 7 to get a generalized Laplacian $L$, and then form the standard Laplacian $M$ by adding an additional row and column $n+1$ with $M_{i,n+1} = L_{ii} - \sum_{j=1}^{n} L_{ij}$ and $M_{n+1,n+1} = \sum_{j=1}^{n} M_{j,n}$. Then $u = Lv$ iff there exists $z$ s.t. $(u, z) = M(v, 0)$ where $(v, 0)$ denotes the vector in $\mathbb{R}^{n+1}$ given by adding final coordinate 0. Furthermore it must be that $\sum_i u_i + z = 0$ because $(u, z)$ lies in the span of $M$. Using the relation between $L$ and $M$ and the triangle inequality and monotonicity (Lemma 5) through the added node $n+1$ we observe

$$
\begin{aligned}
\mathrm{Var}(X_i|X_j) &= \frac{1}{2}(e_i - e_i')^T L^{-1}(e_i - e_i') \\
&= \frac{1}{2}(e_i - e_i')^T M^+(e_i - e_i') \\
&\leq \frac{1}{2}(e_i - e_{n+1})^T M^+(e_i - e_{n+1}) + \frac{1}{2}(e_i' - e_{n+1})^T M^+(e_i' - e_{n+1}) \\
&\leq \frac{1}{2}\frac{1}{M_{i,n+1}} + \frac{1}{2}\frac{1}{M_{i',n+1}} \leq \frac{1}{|\Theta_{ij}|}.
\end{aligned}
$$

$\square$

**Lemma 9.** *Suppose that $i$ is a non-isolated node with $d$ neighbors in an SDD GGM. Then for at least one neighbor $j$ it holds that*

$$
\mathrm{Var}(X_i|X_j) \leq \frac{4d}{\Theta_{ii}}
$$

*Proof.* We establish the following dichotomy: either $\mathrm{Var}(X_i)$ is already small, or if it is large then there is a $j$ s.t. $1/|\Theta_{ij}|$ is small so $\mathrm{Var}(X_i|X_j)$ is small. Observe by Cauchy-Schwartz that

$$
\Theta_{ii}\mathrm{Var}(\mathbb{E}[X_i|X_{\sim i}]) = \Theta_{ii}\mathrm{Cov}(\mathbb{E}[X_i|X_{\sim i}], \mathbb{E}[X_i|X_{\sim i}]) = \sum_j -\Theta_{ij}\mathrm{Cov}(\mathbb{E}[X_i|X_{\sim i}], X_j)
$$

$$
\leq \sum_j |\Theta_{ij}|\sqrt{\mathrm{Var}(\mathbb{E}[X_i|X_{\sim i}])\mathrm{Var}(X_j)}
$$

so

$$
\Theta_{ii}\sqrt{\mathrm{Var}(\mathbb{E}[X_i|X_{\sim i}])} \leq \sum_j |\Theta_{ij}|\sqrt{\mathrm{Var}(X_j)} \leq \sum_j |\Theta_{ij}|\sqrt{\mathrm{Var}(X_i) + 1/|\Theta_{ij}|}
$$

$$
\leq \sqrt{\mathrm{Var}(X_i)}\sum_j |\Theta_{ij}| + \sum_j \sqrt{|\Theta_{ij}|}
$$

$$
\leq \sqrt{\mathrm{Var}(X_i)}\sum_j |\Theta_{ij}| + \sqrt{d\Theta_{ii}}
$$

where in the second inequality we used Lemma 7, in the third inequality we used $\sqrt{a+b} \leq \sqrt{a} + \sqrt{b}$, and in the fourth inequality we used Cauchy-Schwartz and the SDD assumption.

Suppose that $\mathrm{Var}(\mathbb{E}[X_i|X_{\sim i}]) > 4d/\Theta_{ii}$. Then by subtracting $d\sqrt{\Theta_{ii}}$ from both sides we see

$$
\frac{1}{2}\Theta_{ii}\sqrt{\mathrm{Var}(\mathbb{E}[X_i|X_{\sim i}])} \leq \sqrt{\mathrm{Var}(X_i)}\sum_j |\Theta_{ij}| \leq \sqrt{\mathrm{Var}(X_i)}d\max_j |\Theta_{ij}|
$$

so using that $\mathrm{Var}(\mathbb{E}[X_i|X_{\sim i}]) = \mathrm{Var}(X_i) - 1/\Theta_{ii} \geq \mathrm{Var}(X_i)/2$ under our assumption, we find

$$
\frac{\Theta_{ii}}{4d} \leq \frac{\Theta_{ii}}{2d}\sqrt{\frac{\mathrm{Var}(\mathbb{E}[X_i|X_{\sim i}])}{\mathrm{Var}(X_i)}} \leq \max_j |\Theta_{ij}|.
$$

Let $j$ be the maximizer, then from Lemma 8 we find $\mathrm{Var}(X_i|X_j) \leq \frac{1}{|\Theta_{ij}|} \leq \frac{4d}{\Theta_{ii}}$, assuming that $\mathrm{Var}(X_i) > 4d/\Theta_{ii}$. Otherwise, by the law of total variance we know $\mathrm{Var}(X_i|X_j) \leq \mathrm{Var}(X_i) \leq 4d/\Theta_{ii}$. $\square$

**Remark 2** (Electrical intuition for Lemma 9). *We explain the electrical intuition behind Lemma 9 in the case of attractive GGMs. First w.l.o.g. we rescale $\Theta$ to be a generalized Laplacian (Theorem 5). By the electrical interpretation, we think of the edges of the graph are a collection of resistors connecting the nodes, and we imagine connecting the plus end of a 1-volt battery to node $i$, so the effective resistance between the plus and minus end of the battery is the reciprocal of the total current which flows. Then $1/\Theta_{ii}$ is the effective resistance when we connect all of the neighbors of node $i$ directly to the negative end of the battery.*

*When we do this experiment, we know that a lot of the current is either (1) going directly from node $i$ to ground or (2) going from node $i$ to one of its neighbors $j$. In case (1), $\mathrm{Var}(X_i)$ is already small. Otherwise, we are in case (2). In this case, we would expect that if we only grounded node $j$, then the resulting effective resistance $\mathrm{Var}(X_i|X_j)$ should already be quite small; more precisely, within a $O(d)$ factor of grounding all of them, and this is exactly what Lemma 9 says.*

The following example shows that the assumption that the matrix is SDD (or walk-summable) is necessary for the previous Lemmas to be true:

**Example 3** (Failure of Lemma 8 in Non-SDD GGM). *Consider for $\kappa$ fixed and $C$ large*

$$\Theta := \begin{bmatrix} 1 & C & -C \\ C & C^2/\kappa^2 & -C^2/\kappa^2 + 1 \\ -C & -C^2/\kappa^2 + 1 & C^2/\kappa^2 \end{bmatrix}$$

*We can verify that as $C \to \infty$ that the variances (i.e. diagonal of $\Theta^{-1}$) remain $\Theta(1)$ and the matrix is positive definite; furthermore this model is $\kappa$-nondegenerate. However, even after conditioning out the first node, the variance of the second (and third) node remains $\Omega(1) \gg 1/C$.*

## D    Estimating changes in conditional variance

As alluded to before, our algorithms rely on estimating (differences of) conditional variances $\mathrm{Var}(X_i|X_S)$. The classical approach for estimating them is to solve a linear regression problem trying to predict $X_i$ from $X_S$. As we are working in a sample-starved regime and deal with ill-conditioned matrices, we require very fine grained results about such estimates. We collect such results in this section.

For the analysis of Algorithm HYBRIDMB we only need the basic facts from Section 4.1; for the analysis of Algorithm GREEDYANDPRUNE the key additional fact we need is encapsulated as Lemma 14 in Section 4.3 below; finally, for the analysis of the Algorithm SEARCHANDVALIDATE we will also directly use the results stated in Section 4.2.

### D.1    Background: Fixed Design Linear Regression

In this section we recall the standard model for linear regression with Gaussian noise and the usual ordinary least squares estimator and some classical facts about it. See Chapter 14 of [18] for a reference.

**Definition 6** (Fixed design regression with Gaussian noise). *The (well-specified) fixed design regression model is specified by an unknown parameter $w \in \mathbb{R}^k$, known design matrix $\mathbb{X} : m \times k$ with $m > k$ and observations*

$$\mathbb{Y} = \mathbb{X}w + \Xi$$

*where $\Xi \sim N(0, \sigma^2 I)$. In other words, $\mathbb{Y} \sim N(\mathbb{X}w, \sigma^2 I)$.*

**Definition 7** (Ordinary Least Squares (OLS) Estimator). *The OLS estimator for $w$ in the fixed design regression model is the minimizer of*

$$\min_{w} \|\mathbb{Y} - \mathbb{X}w\|_2^2$$

*explicitly given by*

$$\hat{w} := (\mathbb{X}^T \mathbb{X})^{-1} \mathbb{X}^T \mathbb{Y}$$

*assuming that $\mathbb{X}$ has maximal column rank. The corresponding estimator for $\sigma$ is given by*

$$\hat{\sigma}^2 := \frac{1}{m-k} \|\mathbb{Y} - \mathbb{X}\hat{w}\|_2^2.$$

**Fact 2** ([18]). *Under the fixed design regression model with Gaussian noise, $\hat{w} \sim N(w, \sigma^2(\mathbb{X}^T\mathbb{X})^{-1})$ and $\frac{(m-k)\hat{\sigma}^2}{\sigma^2} \sim \chi^2_{m-k}$ where $\chi^2_{m-k}$ denotes a $\chi^2$-distribution with $m-k$ degrees of freedom. Furthermore, $\hat{w}$ and $\hat{\sigma}$ are independent.*

**Lemma 10.** *For any $\delta \in (0, 1)$,*

$$\Pr\left(\left|\frac{\hat{\sigma}^2}{\sigma^2} - 1\right| > 2\sqrt{\frac{\log(2/\delta)}{m-k}} + 2\frac{\log(2/\delta)}{m-k}\right) \leq \delta.$$

*Proof.* Combine Fact 2 and and the concentration inequality from Lemma 4. □

We end with a geometric interpretation of the OLS coordinates which is analogous to Lemma 3. In statistics this is known as the equivalence of the regression $t$-test and the 1-variable regression $F$-test [18].

**Lemma 11.**

$$\min_{w} \|\mathbb{Y} - \mathbb{X}w\|_2^2 - \min_{w:w_i=0} \|\mathbb{Y} - \mathbb{X}w\|_2^2 = \frac{\hat{w}_i^2}{[(\mathbb{X}^T\mathbb{X})^{-1}]_{ii}}$$

*Proof sketch.* Let $\mathbb{X}_i$ be the $i$'th column of $\mathbb{X}$. By the definition of the OLS estimate $\hat{w}$ and the Pythagorean theorem, the left hand side is equal to $\min_{w:w_i=0} \|\mathbb{X}\hat{w} - \mathbb{X}w\|_2^2$. By another application of the Pythagorean theorem, this equals $\|\mathbb{X}_i\hat{w}_i - \mathrm{Proj}_{V_i}\mathbb{X}_i\hat{w}_i\|_2^2 = \hat{w}_i^2\|\mathbb{X}_i - \mathrm{Proj}_{V_i}\mathbb{X}_i\|_2^2$ where $V_i$ is the subspace spanned by the columns of $\mathbb{X}$ except for $i$. Finally $\|\mathbb{X}_i - \mathrm{Proj}_{V_i}\mathbb{X}_i\|_2^2 = \frac{1}{[(\mathbb{X}^T\mathbb{X})^{-1}]_{ii}}$ by applying Schur complement formulas. □

### D.2 Background: Random Design Linear Regression and Wishart Matrices

Under fixed design, the matrix $\mathbb{X}$ was considered to be a deterministic quantity. Random design (see e.g. [56] for references) corresponds to the case where the rows of $\mathbb{X}$ are i.i.d. samples from some distribution, which fits the usual setup in statistical learning theory.

**Definition 8** (Random design linear regression with Gaussian covariates). *The random design linear regression model with Gaussian covariates with $m$ samples is given by a (typically unknown) covariance matrix $\Sigma : k \times k$, i.i.d. samples $X^{(1)}, \ldots, X^{(m)} \sim N(0, \Sigma)$ and corresponding observations*

$$Y^{(i)} = \langle X^{(i)}, w \rangle + \xi^{(i)}, \quad i = 1, \ldots, m \tag{3}$$

*where each $\xi^{(i)} \sim N(0, \sigma^2)$ is independent noise. (The assumption that $\xi^{(i)}$ is independent is referred to as the model being* well-specified.*)*

The OLS estimator is defined as before in Definition 7 where the rows of the design matrix $\mathbb{X}$ are the samples $X_1, \ldots, X_m$ and $\mathbb{Y} = (Y^{(i)})_{i=1}^m$. From (2) we still have that for fixed $X_1, \ldots, X_m$ (i.e. considering only the randomness over $\xi_1, \ldots, \xi_m$)

$$\hat{w}_{OLS} \sim N(w, \sigma^2(\mathbb{X}^T\mathbb{X})^{-1}).$$

Therefore reasoning about the OLS estimator under random design can be reduced to understanding the random matrix $\mathbb{X}^T\mathbb{X}$, which is referred to as a *Wishart matrix* (with $m$ degrees of freedom). We recall here a standard concentration inequality for Wishart matrices when $\Sigma = I$. (This inequality generalizes to the sub-Gaussian case and we have specialized it for simplicity.)

**Theorem 6** (Theorem 4.6.1, [43]). *Suppose that $X^{(1)}, \ldots, X^{(m)} \sim N(0, I)$ are independent Gaussian random vectors in $\mathbb{R}^k$, then*

$$\left\|\frac{1}{m}\sum_{i=1}^m X^{(i)}(X^{(i)})^T - Id\right\| \leq C_1\left(\sqrt{\frac{k}{m}} + \sqrt{\frac{\log(2/\delta)}{m}}\right)$$

*for some absolute constant $C_1 > 0$, with probability at least $1 - \delta$.*

This leads to a multiplicative guarantee for general Wishart matrices:

**Lemma 12.** *Suppose $\epsilon \in (0, 1/2)$ and $\delta > 0$. Then for any $m$ such that $\epsilon \leq C_1 \left( \sqrt{\frac{k}{m}} + \sqrt{\frac{\log(2/\delta)}{m}} \right)$ and $X^{(1)}, \ldots, X^{(m)} \sim N(0, I)$ we have that*

$$(1 - \epsilon)\Sigma \preceq \frac{1}{m} \sum_i X_i X_i^T \preceq (1 + \epsilon)\Sigma$$

*with probability at least $1 - \delta$.*

*Proof.* This is equivalent to showing that

$$(1 - \epsilon)I \preceq \frac{1}{m} \sum_i \Sigma^{-1/2} X^{(i)} (\Sigma^{-1/2} X^{(i)})^T \preceq (1 + \epsilon)I$$

since the PSD ordering is preserved under matrix congruence. The above follows from applying Theorem 6 to $\bar{X}^{(i)} = \Sigma^{-1/2} X^{(i)}$. $\qquad\square$

**Definition 9.** *Given i.i.d. mean-zero random vectors $X^{(1)}, \ldots, X^{(m)}$ the empirical covariance matrix is*

$$\widehat{\Sigma} := \frac{1}{m} \sum_i X^{(i)} (X^{(i)})^T.$$

### D.3 Estimating changes in conditional variance

We are now ready to state what we need for estimating changes in conditional variance. Recall the basic setup: Given samples from $X$ from a GGM at various stages in our algorithm we use estimates for conditional variances of the form $\mathrm{Var}(X_i | X_S)$ by regressing $X_i$ against $X_S$. What we really we need are not actual values of $\mathrm{Var}(X_i | X_S)$ but to find a variable $j \notin S$ that gives non-trivial (or even *most*) advantage in predicting $X_i | X_{S \cup \{j\}}$. So we need to quantify the relative advantage of including an additional variable $j$ on top of $S$.

We can abstract the above in the regression setting as follows: Given samples for regression $(X, Y)$, and an index $j$ check if $\mathrm{Var}(Y|X) = \mathrm{Var}(Y|X_{\sim j})$. That is, whether including feature $x_j$ gives non-trivial advantage in regression. This is akin to the classical *regression t-test* in statistics (see [18]) used to test the null hypothesis that $w_i = 0$ in a linear regression problem.

In the greedy steps in our learning algorithm, we will need to not only find a feature which has a nonzero value for predicting $Y$, but in fact we want to find one of the most predictive features. We do so by exploiting what is known as a *non-central $F$-statistic* [18]. The following lemma quantifies the *usefulness* of a particular coordinate for estimating $Y$. Crucially, this Lemma shows we can estimate the (normalized) change in conditional variance much more accurately than we can actually estimate the individual conditional variances. Note that by Lemma 11 that the term which appears in the Lemma, $\frac{|\hat{w}_j|^2}{(\hat{\Sigma}^{-1})_{jj}}$, also equals the difference in squared loss over the data between the OLS estimator constrained to $w_j = 0$ and the unconstrained OLS estimator.

**Lemma 13.** *Consider the Gaussian random design regression setup (3), fix $j \in \{1, \ldots, k\}$ and let*

$$\gamma := \frac{\mathrm{Var}(Y|X_{\sim j}) - \mathrm{Var}(Y|X)}{\mathrm{Var}(Y|X)}$$

*where $X_{\sim j} = (X_i)_{i \neq j}$. We have*

$$\left| \frac{|\hat{w}_j|}{\hat{\sigma}\sqrt{(\hat{\Sigma}^{-1})_{jj}}} - \sqrt{\gamma} \right| \leq \sqrt{\frac{4 \log(4/\delta)}{m}} + \sqrt{\frac{\gamma}{64}}$$

*and*

$$\left| \frac{|\hat{w}_j|}{\sigma\sqrt{(\hat{\Sigma}^{-1})_{jj}}} - \sqrt{\gamma} \right| \leq \sqrt{\frac{2 \log(4/\delta)}{m}} + \sqrt{\frac{\gamma}{64}}$$

*with probability at least $1 - \delta$ as long as $m \geq m_0 = O(k + \log(4/\delta))$.*

*Proof.* We prove this result directly. Alternatively and essentially equivalently, one could derive a similar result by using classical results in the fixed design regression setting for non-central F-statistics (Theorem 14.11 of [18], see also Section F below) and then analyzing their behavior under random design using matrix concentration.

Recall from Lemma 3 (applied for fixed $X_S$ and then taking expectations) that

$$\mathbb{E}[Y|X] = \mathbb{E}[Y|X_{\sim j}] + \frac{\text{Cov}(Y, X_j|X_{\sim j})}{\text{Var}(X_j|X_{\sim j})}(X_j - \mathbb{E}[X_j|X_{\sim j}])$$

and that

$$\text{Var}(Y|X_{\sim j}) - \text{Var}(Y|X) = \frac{\text{Cov}(Y, X_j|X_{\sim j})^2}{\text{Var}(X_j|X_{\sim j})}$$

so

$$w_j^2 \text{Var}(X_j|X_{\sim j}) = \text{Var}(Y|X_{\sim j}) - \text{Var}(Y|X). \tag{4}$$

i.e. $\frac{w_j^2}{\sigma^2(\Sigma^{-1})_{jj}} = \gamma$. We know that for fixed $X$, over the randomness of $\xi$ we have $\hat{w}_{OLS} \sim N(w, \frac{\sigma^2}{m}\hat{\Sigma}^{-1})$ by Fact 2, so

$$\frac{\hat{w}_j}{\sigma\sqrt{(\hat{\Sigma}^{-1})_{jj}}} \sim N\left(\frac{w_j}{\sigma\sqrt{(\hat{\Sigma}^{-1})_{jj}}}, \frac{1}{m}\right).$$

Using that $(\Sigma^{-1})_{jj} = \frac{1}{\text{Var}(X_j|X_S)}$, $\sigma = \sqrt{\text{Var}(Y|X)}$, and $\gamma = \frac{\text{Var}(Y|X_{\sim j}) - \text{Var}(Y|X)}{\text{Var}(Y|X)}$ and (4) we find

$$\frac{\hat{w}_j}{\sigma\sqrt{(\hat{\Sigma}^{-1})_{jj}}} \sim N\left(\pm\sqrt{\gamma\frac{(\Sigma^{-1})_{jj}}{(\hat{\Sigma}^{-1})_{jj}}}, \frac{1}{m}\right)$$

where the sign is the sign of $w_j$. Applying $||a| - |b|| \le |a - b|$ and the Gaussian tail bound over the randomness of $\hat{w}$ we find

$$\text{Pr}\left(\left|\left|\frac{|\hat{w}_j|}{\sigma\sqrt{(\hat{\Sigma}^{-1})_{jj}}}\right| - \sqrt{\gamma\frac{(\Sigma^{-1})_{jj}}{(\hat{\Sigma}^{-1})_{jj}}}\right| > t\right) \le \text{Pr}\left(\left|\frac{\hat{w}_j}{\sigma\sqrt{(\hat{\Sigma}^{-1})_{jj}}} \mp \sqrt{\gamma\frac{(\Sigma^{-1})_{jj}}{(\hat{\Sigma}^{-1})_{jj}}}\right| > t\right) \le 2e^{-mt^2/2}.$$

Applying Lemma 10 gives

$$\left|\frac{\hat{\sigma}}{\sigma} - 1\right| \le 2\sqrt{\frac{\log(4/\delta)}{m - k - 1}} + 2\frac{\log(4/\delta)}{m - k - 1}$$

with probability at least $1 - \delta/2$. Therefore as long as $m \ge m_1 = O(k + \log(4/\delta))$ we have $\frac{\hat{\sigma}}{\sigma} \in (7/8, 9/8)$. Taking $t = \sqrt{2\log(4/\delta)/m}$ we have

$$\left|\frac{|\hat{w}_j|}{\hat{\sigma}\sqrt{(\hat{\Sigma}^{-1})_{jj}}} - \sqrt{\gamma}\right| \le \sqrt{\frac{\sigma}{\hat{\sigma}}}\left|\frac{|\hat{w}_j|}{\sigma\sqrt{(\hat{\Sigma}^{-1})_{jj}}} - \sqrt{\gamma\frac{\hat{\sigma}}{\sigma}}\right| + \sqrt{\gamma}\left|1 - \sqrt{\frac{(\hat{\Sigma}^{-1})_{jj}}{(\Sigma^{-1})_{jj}}}\right| \le \sqrt{\frac{4\log(4/\delta)}{m}} + \sqrt{\frac{\gamma}{64}}$$

applying Lemma 12 and requiring $m \ge m_2 = O(k + \log(4/\delta))$, with probability at least $1 - \delta$. A simpler variant of this argument gives the result for $\frac{|\hat{w}_j|}{\sigma\sqrt{(\hat{\Sigma}^{-1})_{jj}}}$ as well. $\square$

In our analysis we will often need to estimate multiplicative changes in a quantity of the form $\text{Var}(Y|X_{\sim j}) - V$ (where e.g. $V = \text{Var}(Y|X, X')$ for some $X'$) so we will use the following variant of the previous Lemma:

**Lemma 14.** *Consider the Gaussian random design regression setup* (3)*, fix* $j \in \{1, \dots, k\}$*, let* $V > 0$ *be arbitrary s.t.* $V < \text{Var}(Y|X)$ *and let*

$$\gamma' := \frac{\text{Var}(Y|X_{\sim j}) - \text{Var}(Y|X)}{\text{Var}(Y|X_{\sim j}) - V}$$

*where $X_{\sim j} = (X_i)_{i \neq j}$. We have*

$$\left| \sqrt{\frac{1}{\mathrm{Var}(Y|X_{\sim j}) - V}} \frac{|\hat{w}_j|}{\sqrt{(\hat{\Sigma}^{-1})_{jj}}} - \sqrt{\gamma'} \right| \leq \sqrt{\frac{\mathrm{Var}(Y|X)}{\mathrm{Var}(Y|X_{\sim j}) - V} \cdot \frac{2\log(4/\delta)}{m}} + \sqrt{\frac{\gamma'}{64}}$$

*with probability at least $1 - \delta$ as long as $m \geq m_0 = O(k + \log(4/\delta))$.*

*Proof.* This follows from Lemma 13 after multiplying through in the guarantee by $\sqrt{\gamma'/\gamma}$, using that $\sigma = \sqrt{\mathrm{Var}(Y|X)}$. $\qquad\square$

# E   Learning all attractive GGMs efficiently

**Definition 10.** *We say that a GGM is* attractive *(or* ferromagnetic*) if $\Theta_{ij} \leq 0$ for all $i \neq j$. (This is the same as requiring that $\Theta$ is an $M$-matrix.)*

**Lemma 15.** *If $\Theta$ is the precision matrix of an attractive GGM, then there exists an invertible diagonal matrix $D$ with nonnegative entries such that $D\Theta D$ is a generalized Laplacian.*

*Proof.* This follows immediately from Theorem 5. $\qquad\square$

A particularly important example of an attractive GGM is the *discrete Gaussian free field* — see [23] for a reference to this and the closely related literature on the *continuum Gaussian free field*.

**Definition 11.** *The* discrete Gaussian free field *on a weighted graph $G$ with zero boundary conditions on $S$ is the GGM with $\Theta$ the Laplacian of $G$, after eliminating the rows and columns corresponding to the nodes in $S$.*

Without boundary conditions, the GFF should be translation invariant and so it does not exist as a probability distribution. One can approach the free boundary situation by taking the Laplacian and adding $\epsilon I$ to make it invertible, which gives a learnable model that is arbitrarily poorly conditioned.

**Example 4** (Gaussian simple random walk)**.** *Consider the discrete Gaussian free field on a path of length $n$ with zero boundary condition on the first node. This process is the same as a simple random walk with $N(0,1)$ increments. That is the resulting distribution is of the form $(X_1, \ldots, X_n)$ where $X_i = \sum_{j \leq i} \eta_j$ for independent and identical $\eta_j \sim N(0,1)$. From the GFF perspective, we can think of this as a discretization of Brownian motion (the one-dimensional (continuum) Gaussian free field).*

**Remark 3.** *Every attractive GGM can be realized from a Gaussian Free Field on a weighted graph in the following way: given an attractive GGM, first rescale the coordinates using the above Lemma so that it is a generalized Laplacian. Then, by adding one node to the model we can make the precision matrix into a standard Laplacian on some weighted graph, and conditioning out the added node recovers the original precision matrix.*

Our main theorem of this section is a sample-efficient algorithm for learning attractive GGMs:

**Theorem 7.** *Fix a $\kappa$-nondegenerate attractive GGM. Algorithm* GREEDYANDPRUNE *returns the true neighborhood of every node $i$ with probability at least $1 - \delta$ for $\nu = \kappa^2/\sqrt{32}, K = 64d\log(4/\kappa^2) + 1$ as long as the number of samples $m \geq m_1$ for $m_1 = O((1/\kappa^2)(K\log(n) + \log(4/\delta)))$. The combined run-time (over all nodes) of the algorithm is $O(K^3 mn^2)$.*

Note that the above immediately implies Theorem 1.

As mentioned in the introduction, Algorithm GREEDYANDPRUNE learns the neighborhood of a node by doing greedy forward selection to minimize the conditioned variance, and then doing pruning to remove non-neighbors from the candidate neighborhood. The greedy forward selection step is known in the compressed sensing literature as *Orthogonal Matching Pursuit* (OMP) (see e.g. [57]). We give a description of the OMP algorithm in the general setting of Section D.1 below, along with pseudocode for GREEDYANDPRUNE.

Algorithm ORTHOGONALMATCHINGPURSUIT($T, \mathbb{X}, \mathbb{Y}$):

1. Set $S_0 := \{\}$.
2. For $t$ from 1 to $T$:
   (a) Choose $j$ which minimizes
   $$\min_{w \,:\, \mathrm{supp}(w) \subset S_{t-1} \cup \{j\}} \|\mathbb{Y} - \mathbb{X}w\|_2^2$$
   (b) Set $S_t := S_{t-1} \cup \{j\}$
3. Return $S_T$.

---

Algorithm GREEDYANDPRUNE($i, \nu, T$):

1. Run ORTHOGONALMATCHINGPURSUIT for $T$ steps to predict $\mathbb{X}_i$ from the other columns of $\mathbb{X}$, i.e. setting $\mathbb{Y} = \mathbb{X}_i$ and $\mathbb{X}' = \mathbb{X}_{\sim i}$.
2. Define $\hat{\Theta}_{ii}$ by $1/\hat{\Theta}_{ii} = \widehat{\mathrm{Var}}(X_i|X_S)$.
3. For $j \in S$:
   (a) Let $S' := S \setminus \{j\}$ and $\hat{w} := \hat{w}(i, S')$.
   (b) If $\widehat{\mathrm{Var}}(X_i|X_{S'}) - \widehat{\mathrm{Var}}(X_i|X_S) < \nu/\hat{\Theta}_{ii}$, set $S := S'$.
4. Return $S$.

**Remark 4** (Implementation: Merging neighborhoods). *In order to return an actual estimate for the inverse precision matrix, we add in our implementation of* GREEDYANDPRUNE *a merging step which includes an edge* $(i, j)$ *iff it is in the computed neighborhood of node* $i$ *and in the computed neighborhood of node* $j$. *Then to estimate the entries, we use OLS to predict node* $X_i$ *from its neighbors and estimate the conditional variance of* $X_i$. *We define* $\hat{\Theta}_{ii}$ *to be the inverse of the estimated conditional variance, and* $-\hat{\Theta}_{ij}/\hat{\Theta}_{ii}$ *to be the OLS coefficient. Finally, we symmetrize* $\hat{\Theta}$ *by picking the smaller of absolute norm between* $\hat{\Theta}_{ij}$ *and* $\hat{\Theta}_{ji}$*; the same step is used in CLIME [58].*

## E.1 Proof of supermodularity

As a first step toward proving Theorem 7, we first show that the conditional variance function is supermodular.

**Definition 12.** *Given a universe U, a function* $f : 2^U \to \mathbb{R}$ *is supermodular if for any* $S \subset T$,
$$f(S) - f(S \cup \{j\}) \geq f(T) - f(T \cup \{j\}).$$
*(This is the same as saying* $-f$ *is submodular.)*

Supermodularity of the conditional variance of a node in the GFF (and hence, by using the reduction from Remark 3, all attractive GGMs) was previously shown independently in [26, 30] using two different methods. The proof in [26] is algebraic using the Schur complement formula, whereas the proof in [30] converts the problem into one about electrical flows and argues via Thomson's principle. We give a third different proof which has the benefit of being transparent and using only basic linear algebra.

**Theorem 8.** *For any node* $i$ *in a ferromagnetic GGM,* $\mathrm{Var}(X_i|X_S)$ *is a monotonically decreasing, supermodular function of* $S$.

*Proof.* By rescaling we may assume w.l.o.g. that $\Theta_{ii} = 1$ for all $i$. Define $\Theta_S$ to be the precision matrix corresponding to conditioning $S$ out (i.e. $\Theta$ with the rows and columns of $S$ removed), and $\Sigma_S = \Theta_S^{-1}$. Then, if we write $\Theta_S = I - A_S$, by Neumann series formula (as $\Theta_S \succ 0$, $\|A_S\| < 1$ using Perron-Frobenius), we see

$$\Sigma_S = (I - A_S)^{-1} = \sum_{k=0}^{\infty} A_S^k. \tag{5}$$

Writing this out explicitly for $(\Sigma_S)_{i,i}$ gives

$$\text{Var}(X_i|X_S) = \sum_{k=0}^{\infty} \sum_{v_1,\dots,v_k \notin S} (-\Theta_{iv_1}) \cdots (-\Theta_{v_k i}), \tag{6}$$

where the $k = 0$ term in the sum is interpreted to be 1, so $\text{Var}(X_i|X_S)$ is a nonnegative weighted sum over walks avoiding $S$ and returning to $i$ in the final step. The above expression is clearly monotonically increasing in $S$ as all off-diagonal entries of $\Theta$ are negative (and also follows from law of total variance); to verify supermodularity, we just need to check that

$$\text{Var}(X_i|X_S) - \text{Var}(X_i|X_{S \cup \{j\}}) = \sum_{k=0}^{\infty} \sum_{\substack{v_1,\dots,v_k \notin S, \\ j \in \{v_1,\dots,v_k\}}} (-\Theta_{iv_1}) \cdots (-\Theta_{v_k i})$$

is a monotonically decreasing function of $S \subseteq [n] \setminus \{i,j\}$, but this is clear once we apply (6) as the set of cycles that are eliminated from the sum by adding $j$ only shrinks as we increase $S$. $\qquad\square$

Supermodularity of the conditional variance has the following consequence which will later be useful in showing that the greedy algorithm makes non-trivial progress in each step.

**Lemma 16.** *For any node $i$ in a ferromagnetic GGM, if $S$ is a set of nodes that does not contain $i$ or all neighbors of $i$, and $T$ is the set of neighbors of $i$ not in $S$, then there exists some node $j \in T$ such that*

$$\text{Var}(X_i|X_S) - \text{Var}(X_i|X_{S \cup \{j\}}) \geq \frac{\text{Var}(X_i|X_S) - 1/\Theta_{ii}}{|T|} .$$

*Proof.* This is a standard consequence of supermodularity – we include the proof for completeness.

Consider adjoining the elements of $T$ to $S$ one at a time, and then apply supermodularity to show

$$\text{Var}(X_i|X_S) - \text{Var}(X_i|X_{S \cup T}) \leq \sum_{j \in T}(\text{Var}(X_i|X_S) - \text{Var}(X_i|X_{S \cup \{j\}}))$$
$$\leq |T| \max_{j \in T}(\text{Var}(X_i|X_S) - \text{Var}(X_i|X_{S \cup \{j\}})).$$

Rearranging and using $\text{Var}(X_i|X_{S \cup T}) = 1/\Theta_{ii}$ (by the Markov property) gives the result. $\qquad\square$

From (5) we see immediately that the entries of the covariance $\Sigma$ of an attractive GGM are always nonnegative (this is why they are called attractive/ferromagnetic); we record this fact for future use.

**Lemma 17** (Griffith's inequality). *In an attractive GGM, $\text{Cov}(X_i, X_j) \geq 0$ for any $i, j$.*

This fact is very well-known, holds for arbitrary ferromagnetic graphical models (i.e. not just Gaussian) and is referred to as *Griffith's inequality*. See [59] for a more general proof.

### E.2 Greedy Subset Selection in Attractive Models

In this section we give a guarantee for *subset selection* using OMP, by showing that after a small number of rounds OMP finds a set $S$ such that $\text{Var}(X_i|X_S)$ is close to minimal. The sample complexity analysis is complicated by the fact that supermodularity holds at the level of the population loss (i.e. for an infinite amount of data) whereas it would be more convenient if it held for the empirical conditional variance, so we have to deal with both the regression noise and the randomness of the regressors. First we prove the following lemma which gives a stronger version of Lemma 2 for ferromagnetic GGMs:

**Lemma 18.** *Fix $i$ a node in a $\kappa$-nondegenerate ferromagnetic GGM, and let $S$ be set of nodes and let $T$ be the set of neighbors of $i$ not in $S$. Then*

$$\text{Var}(X_i|X_S) \geq \frac{1 + |T|\kappa^2}{\Theta_{ii}}$$

*Proof.* By the law of total variance, Griffith's inequality (Lemma 17), and the law of total variance again

$$\mathrm{Var}(X_i|X_S) - \frac{1}{\Theta_{ii}} = \mathrm{Var}(\mathbb{E}[X_i|X_{\sim i}]|X_S) = \mathrm{Var}(\sum_{j \in T} \frac{-\Theta_{ij}}{\Theta_{ii}} X_j | X_S)$$

$$\geq \sum_{j \in T} \frac{\Theta_{ij}^2}{\Theta_{ii}^2} \mathrm{Var}(X_j|X_S) \geq \frac{1}{\Theta_{ii}} \sum_{j \in T} \frac{\Theta_{ij}^2}{\Theta_{ii}\Theta_{jj}} \geq \frac{|T|\kappa^2}{\Theta_{ii}}.$$

$\square$

**Lemma 19.** *Suppose that $X$ is distributed according to an $\kappa$-nondegenerate ferromagnetic GGM and $i$ is a node of degree at most $d$. Let $\sigma^2 := \frac{1}{\Theta_{ii}}$ and $w_j^* = \frac{-\Theta_{ij}}{\Theta_{ii}}$ for all $j \neq i$. Then using $T$ rounds of OMP to predict $X_i$ given $X_{\sim i}$ from $m$ i.i.d. samples, we have that $\mathrm{Var}(\mathbb{E}[X_i|X_{\sim i}]|X_S) \leq (1 - 1/2d)^{T-1}\frac{8d}{\Theta_{ii}}$ with probability at least $1 - \delta$ provided that $m = \Omega((d + 1/\kappa^2)(T\log(n) + \log(2/\delta)))$.*

*Proof.* We prove by induction that for every $1 \leq t \leq T$ that

$$\mathrm{Var}(\mathbb{E}[X_i|X_{\sim i}]|X_{S_t}) \leq (1 - 1/2d)^{t-1}\frac{8d}{\Theta_{ii}}.$$

Note that by Lemma 9 there exists a node $j$ such that $\mathrm{Var}(X_i|X_j) \leq \frac{4d}{\Theta_{ii}}$. By taking a union bound, we may assume that:

1. $\mathrm{Var}(X_i|X_{S_1}) \leq \frac{8d}{\Theta_{ii}}$ using the above fact combined with Lemma 10 assuming that $m = \Omega(\log(n/\delta))$ to guarantee that the estimated conditional variances have small multiplicative error.

2. For all subsets $U$ of $[n]$ of size at most $T$ and $j \in [n]$, applying Lemma 14 we have

$$\left| \frac{1}{\sqrt{\mathrm{Var}(X_i|X_{U\setminus\{j\}}) - 1/\Theta_{ii}}} \hat{R}(U, j) - \sqrt{\gamma'} \right| \leq \sqrt{\frac{\mathrm{Var}(X_i|X_U)}{\mathrm{Var}(X_i|X_{U\setminus\{j\}}) - 1/\Theta_{ii}}} \sqrt{\frac{4(T\log(n) + \log(12/\delta))}{m}} + \sqrt{\frac{\gamma'}{64}}$$

   where

$$\gamma' = \gamma'(U, j) := \frac{\mathrm{Var}(X_i|X_{U\setminus\{j\}}) - \mathrm{Var}(X_i|X_U)}{\mathrm{Var}(X_i|X_{U\setminus\{j\}}) - 1/\Theta_{ii}}$$

   and

$$\hat{R}(U, j) := \frac{(\hat{w}_U)_j}{((\hat{\Sigma}_{U,U})^{-1})_{jj}} = \sqrt{\|\mathbb{X}_i - \mathbb{X}\hat{w}_U\|_2^2 - \|\mathbb{X}_i - \mathbb{X}\hat{w}_{U\setminus\{j\}}\|_2^2}$$

   using Lemma 11 in the last equality where $\hat{w}_U$ is the OLS estimate using only the coordinates in $U$. This holds assuming that $m = \Omega(T\log(4n) + \log(1/\delta))$.

Before proceeding, we observe that

$$\sqrt{\frac{\mathrm{Var}(X_i|X_U)}{\mathrm{Var}(X_i|X_{U\setminus\{j\}}) - 1/\Theta_{ii}}} \leq \sqrt{\frac{\mathrm{Var}(X_i|X_{U\setminus\{j\}})}{\mathrm{Var}(X_i|X_{U\setminus\{j\}}) - 1/\Theta_{ii}}} \leq \max(\sqrt{2}, \sqrt{2/d'\kappa^2}) \quad (7)$$

where $d'$ is the degree of node $i$ in the graph with the nodes in $U \setminus \{j\}$ removed, by the law of total variance (first inequality) and the following case analysis: either $\mathrm{Var}(X_i|X_{U\setminus\{j\}}) \geq 2/\Theta_{ii}$, in which case $\frac{\mathrm{Var}(X_i|X_{U\setminus\{j\}})}{\mathrm{Var}(X_i|X_{U\setminus\{j\}}) - 1/\Theta_{ii}} \leq 2$, or $\mathrm{Var}(X_i|X_{U\setminus\{j\}}) \leq 2/\Theta_{ii}$ in which case $\frac{\mathrm{Var}(X_i|X_{U\setminus\{j\}})}{\mathrm{Var}(X_i|X_{U\setminus\{j\}}) - 1/\Theta_{ii}} \leq 2/d'\kappa^2$ by Lemma 18.

The first point above gives the base case for the induction. By Lemma 16, if $\mathrm{Var}(\mathbb{E}[X_i|X_{\sim i}]|S_t) \neq 0$ then there exists a $k$ such that

$$\gamma'(S_t \cup \{k\}, k) = \frac{\mathrm{Var}(\mathbb{E}[X_i|X_{\sim i}]|X_{S_t}) - \mathrm{Var}(\mathbb{E}[X_i|X_{\sim i}]|X_{S_t \cup \{k\}})}{\mathrm{Var}(\mathbb{E}[X_i|X_{\sim i}]|X_{S_t \cup \{k\}})} \geq \frac{1}{d'}$$

where (as above) $d'$ is the degree of $i$ in the set of non-neighbors of $S_t$. Combined with (7) and $d' \leq d$ we now see that the second guarantee above ensures that at every time $t$, the $j$ selected by OMP (i.e. $j$ where $S_{t+1} = S_t \cup \{j\}$) satisfies $\gamma'(S_t \cup \{j\}, j) \geq 1/2d$ as long as $m = \Omega((d + 1/\kappa^2)(T \log(n) + \log(12/\delta)))$. We therefore have that

$$\mathrm{Var}(X_i | X_{S_t}) - 1/\Theta_{ii} \leq (1 - 1/2d)(\mathrm{Var}(X_i | X_{S_{t-1}}) - 1/\Theta_{ii})$$

for all $1 < t \leq T$, which combined with the induction hypothesis gives the result (using that $\mathrm{Var}(X_i | X_{S_t}) - 1/\Theta_{ii} = \mathrm{Var}(\mathbb{E}[X_i | X_{\sim i}] | X_{S_t})$ by law of total variance). $\qquad \square$

### E.3 Structure Recovery for Attractive GGMs

To give a final result for structure recovery, we show how to combine the previous analysis of greedy forward selection with a simple analysis of pruning (backward selection).

**Lemma 20.** *Let $i$ be a node of degree at most $d$ in a $\kappa$-nondegenerate attractive GGM. Fix $\delta > 0$ and suppose that $m = \Omega((d + 1/\kappa^2)(T \log(n) + \log(2/\delta)))$ where $T = \Theta(d \log(2d/\kappa^2))$. Then with probability at least $1 - \delta$, the neighborhod of node $i$ is correctly recovered by Algorithm GREEDYANDPRUNE with $\nu = \Theta(\kappa^2)$.*

*Proof.* By Lemma 19 with $T = 1 + 2d \log(16d/\kappa^2)$, with probability at least $1 - \delta/2$ we have that $\mathrm{Var}(\mathbb{E}[X_i | X_{\sim i}] \mid X_S) \leq \kappa^2/2$ where $S$ is the set returned by OMP as long as $m = \Omega((d + 1/\kappa^2)(T \log(n) + \log(24/\delta)))$. From Lemma 2 we see this implies that $S$ contains the true neighborhood of node $i$.

We now analyze the pruning step for any $S$ which is a superset of the true neighborhood of size at most $T$. By Lemma 2 and the Markov property, we know that if $j$ is a true neighbor then $\gamma(S, j) \geq \kappa^2$, and otherwise $\gamma(S, j) = 0$. Applying Lemma 13 and taking the union bound over the at most $n^T$ possible sets, we find that exactly the true edges are kept with probability at least $1 - \delta/2$ as long as $m = \Omega((T \log(n) + \log(8/\delta))/\kappa^2)$. Therefore the entire neighborhood recovery succeeds with probability at least $1 - \delta$. $\qquad \square$

**Theorem 9.** *Let $X$ be distributed according to a $\kappa$-nondegenerate GGM on $n$ nodes with maximum degree $d$. Fix $\delta > 0$, then with probability at least $1 - \delta$ Algorithm GREEDYANDPRUNE run at every node with $T = \Theta(d \log(2d/\kappa^2))$ and $\nu = \Theta(\kappa^2)$ successfully recovers the true graph as long as $m = \Omega((1/\kappa^2)(d \log(2d/\kappa^2) + \log(2/\delta)) \log(n))$.*

*Proof.* This follows from Lemma 20 by taking the union bound over the $n$ nodes and recalling from Lemma 6 the bound $d \leq 1/\kappa^2$. $\qquad \square$

**Remark 5** (Input specification)**.** *In the description of the algorithms throughout this paper, we assume we have access to i.i.d. samples from the distribution. However, it is straightforward to verify that the algorithms only depend on the empirical covariance matrix, and can be run given only the empirical covariance matrix in polynomial time.*

## F Information-theoretic optimal learning of attractive GGMs

In this section we give an $O(n^d)$ time algorithm for recovering attractive GGMs which matches the information-theoretic lower bounds up to constants, improving the result of the previous section at the cost of computational efficiency.

### F.1 Noncentral F-statistics

In the analysis of the $O(n^d)$ time algorithm, we will need to compare empirical variances between predictors supported on very different sets of variables. In comparison, in the analysis of greedy methods we only needed to consider adding or removing a single variable at a time. In order to handle the new setting, we recall the definition of noncentral F-statistics and their connection to fixed design regression.

**Definition 13.** *Suppose $Z_1 \sim N(\delta, 1)$ and for $j > 1$, $Z_j \sim N(0, 1)$ with $Z_1, \ldots, Z_m$ independent. Then we write $\sum_i Z_i \sim \chi_m^2(\delta^2)$ where $\chi_m^2(\delta^2)$ is the* noncentral chi-square distribution *with noncentrality parameter $\delta^2$ and $m$ degrees of freedom.*

**Definition 14.** *If $V \sim \chi_k^2(\delta^2)$ and $W \sim \chi_m^2$ is independent of $V$, then we write*

$$\frac{V/k}{W/m} \sim F_{k,m}(\delta^2)$$

*where $F_{k,m}(\delta^2)$ is the* noncentral F-distribution *with degrees of freedom $k$ and $m$ and noncentrality parameter $\delta^2$.*

**Theorem 10** (Theorem 14.11 of [18])**.** *In the (Gaussian) fixed design regression model (Section D.1), let $H$ be a $q$-dimensional subspace of $\mathbb{R}^k$. Define*

$$T := \frac{m-k}{k-q} \frac{\|\mathbb{Y} - \mathbb{X}\hat{w}_0\|^2 - \|\mathbb{Y} - \mathbb{X}\hat{w}\|^2}{\|\mathbb{Y} - \mathbb{X}\hat{w}\|^2} = \frac{m-k}{k-q} \frac{\|\mathbb{X}\hat{w} - \mathbb{X}\hat{w}_0\|^2}{\|\mathbb{Y} - \mathbb{X}\hat{w}\|^2}$$

*where $\hat{w}$ is the unrestricted OLS estimator and $\hat{w}_0$ is the least squares estimator constrained to be inside of subspace $H$. (The second equality holds by the Pythagorean theorem.) Then $T \sim F_{k-q,m-k}(\gamma)$ where*

$$\gamma := \frac{\min_{w_0 \in H_0} \|\mathbb{X}(w - w_0)\|^2}{\sigma^2}.$$

*More specifically, $\frac{1}{\sigma^2}\|\mathbb{Y} - \mathbb{X}\hat{w}\|^2 \sim \chi_{m-k}^2$ and $\frac{1}{\sigma^2}\|\mathbb{X}\hat{w} - \mathbb{X}\hat{w}_0\|^2 \sim \chi_{k-q}^2(\gamma)$ and these random variables are independent.*

We also recall a convenient concentration inequality for noncentral $\chi^2$-distributed random variables:

**Lemma 21** (Lemma 8.1 of [60])**.** *Suppose that $V \sim \chi_m^2(\delta^2)$. Then*

$$\Pr(V \geq (m + \delta^2) + 2\sqrt{(m + 2\delta^2)t} + 2t) \leq e^{-t}$$

*and*

$$\Pr(V \leq (m + \delta^2) - 2\sqrt{(m + 2\delta^2)t}) \leq e^{-t}.$$

## F.2 Structure learning by $\ell_0$-constrained least squares

We perform structure recovery by, for every node $i$, performing several $\ell_0$-constrained regressions and pruning the result. In the context of learning Gaussian graphical models, some algorithms in a similar spirit referred to as SLICE and DICE were proposed in [14] and they proved a sample complexity bound of $O(d/\kappa^2 \log(n))$ for the more sample-efficient method, DICE. We show our estimator SEARCHANDVALIDATE actually achieves optimal sample complexity $O((1/\kappa^2)\log(n))$ in the setting of attractive GGMs, and always achieves a sample complexity of $O((d/\kappa^2)\log(n))$ which gives a faster algorithm with the same sample complexity as DICE from [14], which has a slower runtime of $O(n^{2d+1})$. (It matches the runtime guarantee for SLICE in [14], which has a worse sample complexity guarantee.)

In Algorithm SEARCHANDVALIDATE, the key step is performing $\ell_0$-constrained regression to predict $X_i$; the loop in step 2 is required only because we do not know a priori the exact degree of node $i$, only an upper bound. With high probability, the support of one of the $w_{d'}$ will equal the exact neighborhood of node $i$, and then a straightforward validation procedure in step 3 (which uses a similar idea to Algorithm DICE in [14]) allows us to identify the correct $w_{d'}$ successfully. For the purposes of the analysis, for every pair of sets $S_0 \subset S$ not containing $i$ define (as in Theorem 10)

$$T(S_0, S) := \frac{n - |S|}{|S| - |S_0|} \frac{\|\mathbb{X}_i - \mathbb{X}\hat{w}_0\|^2 - \|\mathbb{X}_i - \mathbb{X}\hat{w}\|^2}{\|\mathbb{X}_i - \mathbb{X}\hat{w}\|^2} = \frac{n - |S|}{|S| - |S_0|} \frac{\|\mathbb{X}\hat{w} - \mathbb{X}\hat{w}_0\|^2}{\|\mathbb{X}_i - \mathbb{X}\hat{w}\|^2}$$

where $\hat{w}_0$ is the OLS estimator restricted to $\operatorname{supp}(w_0) \subset S_0$ and $\hat{w}$ is the OLS estimator restricted to $\operatorname{supp}(w) \subset S$.

The following Lemma analyzes the key step in the above algorithm; it shows that when $d'$ equals the true degree of node $i$, the true support is returned. The crucial part which requires that the GGM is attractive is the application of Lemma 18, which guarantees that candidate supports which are far away from the true neighborhood perform much worse than the true neighborhood. This is crucial because there are many candidate neighborhoods far away from the true neighborhood, which means we need an improved bound to handle them and overcome the cost of taking the union bound.

Algorithm SEARCHANDVALIDATE(i,d,$\nu$):

1. We assume the data has been split into two equally sized sample sets 1 and 2. Let $\hat{\mathbb{E}}_1$ and $\hat{\mathbb{E}}_2$ denote the empirical expectation over these two sets and define $\widehat{\mathrm{Var}}_2$ similarly.

2. For $d'$ in 0 to $d$:

   (a) Find $w_{d'}$ minimizing

   $$\min_{w:w_i=0,|\operatorname{supp}(w)|\leq d'}\hat{\mathbb{E}}_1[(X_i - w_{d'}\cdot X)^2]$$

3. For $d'$ in 0 to $d$ (outer loop):

   (a) For $d''$ in 0 to $d$ except $d'$ (inner loop):

      i. Let $S_{d',d''} := \operatorname{supp}(w_{d'}) \cup \operatorname{supp}(w_{d''})$.

      ii. For $j$ in $\operatorname{supp}(w_{d''}) \setminus \operatorname{supp}(w_{d'})$

         A. If $\widehat{\mathrm{Var}}_2(X_i|X_{S_{d',d''}\setminus\{j\}}) - \widehat{\mathrm{Var}}_2(X_i|X_{S_{d',d''}}) > \nu\widehat{\mathrm{Var}}_2(X_i|X_{S_{d',d''}})$, continue to next iteration of outer loop.

   (b) Return $\operatorname{supp}(w_{d'})$.

**Lemma 22.** *In a $\kappa$-nondegenerate attractive GGM, if $i$ is a node of degree $d$ then $\ell_0$ constrained regression over vectors with support size at most $d$ returns the true neighborhood of node $i$ with probability at least $1 - \delta$ as long as $m = \Omega(\log(n)/\kappa^2 + \log(2/\delta)/\kappa^2)$.*

*Proof.* First we consider the randomness over the samples of $X_{\sim i}$, i.e. over $\mathbb{X}$ with column $i$ removed. By Lemma 12 and the union bound over all subsets $S$ of $[n] \setminus \{i\}$ with $|S| \leq 2d$, it holds with probability at least $1 - \delta/2$ that for all $w$ with $w_i = 0$ and $|\operatorname{supp}(w)| \leq 2d$,

$$\frac{1}{2}\mathbb{E}[(w^TX)^2] \leq \frac{1}{2}w^T\left(\frac{1}{m}\mathbb{X}^T\mathbb{X}\right)w \leq \mathbb{E}[(w^TX)^2] \tag{8}$$

as long as $m = \Omega(d\log(n) + \log(2/\delta))$. (Recall from Lemma 6 that $d \leq 1/\kappa^2$, so this holds under the hypothesis of the theorem.) We condition on this event and consider the remaining randomness over $\mathbb{X}_i$. Let $S^*$ be the set of true neighbors of node $i$ and let $S_0$ be any other subset of size at most $d$. Define $S := S^* \cup S_0$. Since the OLS estimators are defined by projection onto spans of the columns of $\mathbb{X}$, we can apply the Pythagorean theorem to get

$$\|\mathbb{X}_i - \mathbb{X}\hat{w}_{S^*}\|^2 = \|\mathbb{X}_i - \mathbb{X}w_S\|^2 + \|\mathbb{X}\hat{w}_{S^*} - \mathbb{X}\hat{w}_S\|^2$$

and

$$\|\mathbb{X}_i - \mathbb{X}\hat{w}_{S_0}\|^2 = \|\mathbb{X}_i - \mathbb{X}w_S\|^2 + \|\mathbb{X}\hat{w}_{S_0} - \mathbb{X}\hat{w}_S\|^2.$$

Subtracting, we get that

$$\|\mathbb{X}_i - \mathbb{X}\hat{w}_{S_0}\|^2 - \|\mathbb{X}_i - \mathbb{X}\hat{w}_{S^*}\|^2 = \|\mathbb{X}\hat{w}_{S_0} - \mathbb{X}\hat{w}_S\|^2 - \|\mathbb{X}\hat{w}_{S^*} - \mathbb{X}\hat{w}_S\|^2.$$

To prove the result, it suffices to show with high probability that for any $S_0$ which does not contain $S^*$ that the leftmost term is positive — then no such $S_0$ can be the minimizer of the $\ell_0$-constrained regression, since $S^*$ corresponds to a feasible point with smaller objective value. We achieve this by showing the right hand side is positive. Observe

$$\|\mathbb{X}\hat{w}_{S_0} - \mathbb{X}\hat{w}_S\|^2 - \|\mathbb{X}\hat{w}_{S^*} - \mathbb{X}\hat{w}_S\|^2 = \frac{d-q}{n-|S|}\|\mathbb{Y} - \mathbb{X}\hat{w}_S\|^2(T(S_0,S) - T(S^*,S)).$$

where $q = |S_0| = |S^*|$ so it suffices to show that $T(S_0,S) - T(S^*,S) \geq 0$. In fact, canceling out denominators, dividing by $\sigma^2$ and rearranging it suffices to show

$$\frac{1}{\sigma^2}\|\mathbb{X}\hat{w}_S - \mathbb{X}\hat{w}_{S_0}\|^2 \geq \frac{1}{\sigma^2}\|\mathbb{X}\hat{w}_S - \mathbb{X}\hat{w}_{S^*}\|^2$$

where by Theorem 10 the left hand side is according to $\chi^2_{d-q}(\gamma)$ with $\gamma := \frac{\min_{\operatorname{supp}(w_0)\subset S}\|\mathbb{X}(w_0-w^*)\|^2}{\sigma^2}$ and the right hand side is distributed according to $\chi^2_{d-q}$, where

$\sigma^2 := 1/\Theta_{ii}$. Observe by (8) that

$$\gamma \geq \frac{m}{2} \frac{\min_{\mathrm{supp}(w_0) \subset S} \mathbb{E}[(X^T(w_0 - w^*))^2]}{\sigma^2} = \frac{m}{2} \frac{\min_{\mathrm{supp}(w_0) \subset S} \mathrm{Var}(X^T(w_0 - w^*))]}{\sigma^2} \geq \frac{m\kappa^2(d-q)}{2} \tag{9}$$

where the last inequality is by Lemma 18, since $w_0$ is supported on $S_0$ which is missing $d - q$ of the neighbors of node $i$. Applying Lemma 21

$$\Pr(\frac{1}{\sigma^2}\|\mathbb{X}\hat{w}_S - \mathbb{X}\hat{w}_{S_0}\|^2 \leq (d - q + \gamma) - 2\sqrt{(d-q+2\gamma)t}) \leq e^{-t}$$

and applying Lemma 4

$$\Pr(\frac{1}{\sigma^2}\|\mathbb{X}\hat{w}_S - \mathbb{X}\hat{w}_{S^*}\|^2 \geq (d-q) + 2\sqrt{(d-q)t} + 2t) \leq e^{-t}.$$

Letting $t = \log(4dn^{d-q}/\delta)$, and taking the union bound over the at most $n^{d-q}$ possible values of $S_0$ and then over the at most $d$ possible values of $q$, we find that with probability at least $1 - \delta/2$ for all possible $S_0$ and $q$ that

$$\frac{1}{\sigma^2}\|\mathbb{X}\hat{w}_S - \mathbb{X}\hat{w}_{S_0}\|^2 - \frac{1}{\sigma^2}\|\mathbb{X}\hat{w}_S - \mathbb{X}\hat{w}_{S^*}\|^2 \geq \gamma - 2\sqrt{(d-q+2\gamma)t} - 2\sqrt{(d-q)t} \geq \gamma - 4\sqrt{(d-q+2\gamma)t}.$$

Finally, we see this is nonnegative as long as $\gamma = \Omega(t) = \Omega((d-q)\log(n) + \log(2/\delta))$, which by (9) holds as long as $m = \Omega(\frac{\log(n) + \log(2/\delta)}{\kappa^2})$. Therefore the desired result holds with total probability at least $1 - \delta$, completing the proof. $\qquad\square$

**Theorem 11.** *Fix $\delta > 0$. In a $\kappa$-nondegenerate attractive GGM, as long as $m = \Omega((1/\kappa^2)\log(n) + \log(2/\delta)/\kappa^2)$ it holds with probability at least $1 - \delta$ that Algorithm* SEARCHANDVALIDATE *with $\nu = \kappa^2/2$ returns the true neighborhood of every node $i$.*

*Proof.* By applying Lemma 22 and taking the union bound over nodes $i$, we know that as long as $m = \Omega((1/\kappa^2)\log(n) + \log(2/\delta)/\kappa^2)$ then with probability at least $1 - \delta/2$ for every node $i$, for $d'$ equal to the true degree of node $i$ that $w_{d'}$ returned in step 2 of Algorithm SEARCHANDVALIDATE is supported on exactly the true neighborhood of node $i$.

Furthermore, conditioned on the previous event (which only involves sample set 1), it holds with probability at least $1 - \delta/2$ by taking the union bound over the possible values of $d', d''$ that (similar to the pruning argument used in analysis of Algorithm GREEDYANDPRUNE):

1. in step 3(a).ii, for every $d'$ less than the true degree of node $i$ and for $d''$ equal to the true degree of node $i$ that the outer loop continues to the next step by applying Lemma 13, Lemma 11, and Lemma 2 and considering any $j$ in the true neighborhood and missing from the support of $w_{d'}$.

2. In step 3 when $d'$ equals the true degree of node $i$, step 3(b) is reached and the true support of node $i$ is returned by applying Lemma 13 and Lemma 11.

as long as $m = \Omega((d + 1/\kappa^2)\log(n) + \log(2/\delta)/\kappa^2)$. Using that $d \leq 1/\kappa^2$ by Lemma 6, we see the requirement on $m$ holds and as desired, the algorithm succeeds with total probability at least $1 - \delta$. $\qquad\square$

A simplified argument in the general (non-attractive) case, using the weaker bound from Lemma 2 instead of Lemma 18, yields the following result in the general case.

**Theorem 12.** *Fix $\delta > 0$. In a $\kappa$-nondegenerate (not necessarily attractive) GGM with maximum degree d, as long as $m = \Omega((d/\kappa^2)\log(n) + \log(2/\delta)/\kappa^2)$ it holds with probability at least $1 - \delta$ that Algorithm* SEARCHANDVALIDATE *with $\nu = \kappa^2/2$ returns the true neighborhood of every node $i$.*

# G  Hybrid $\ell_1$ regression guarantees

In the next section, we will discuss algorithms for regression and structure learning in general walk-summable models. Since (as we will see) the conditional variance is not supermodular in these models, we need some fundamentally new tools to analyze this setting. It turns out that we will need to analyze a variant of $\ell_1$-constrained least squares regression, which we do in this section as preparation.

**Definition 15.** *We define the* hybrid $\ell_1$-regression model *to be given by*

$$Y = \langle w^*, X - \mathbb{E}[X|Z] \rangle + a^* Z + \xi$$

*where* $\|w\|_1 \leq W$ *and conditioned on* $Z$, $X - \mathbb{E}[X|Z] \sim N(0, \Sigma)$ *with* $\Sigma : n \times n$, $\Sigma_{ii} \leq R^2$ *for all* $i$, $\mathbb{E}Z^2 = 1$ *(w.l.o.g.), and* $\mathbb{E}\xi^2 = \sigma^2$ *with the noise* $\xi$ *independent of* $X, Z$.

The corresponding function class is

$$\mathcal{F} := \{(x,z) \mapsto \langle w, x - \mathbb{E}[X|Z=z] \rangle + az : \|w\|_1 \leq W\} = \{(x,z) \mapsto \langle w, x \rangle + a'z : \|w\|_1 \leq W\}.$$

and the *Empirical Risk Minimizer* (ERM) is given by taking the minimizer of

$$\min_{\|w\|_1 \leq W, a'} \hat{\mathbb{E}}[(Y - \langle w, X \rangle - a'Z)^2].$$

As mentioned in the introduction, it will be crucial in the analysis to look at the parameterization with $a$ instead of $a'$ even though algorithmically the ERM will be computed using the variable $a'$ (as the change of basis given by subtracting off the conditional expectations is unknown and could only be approximated from data).

## G.1  Guarantees for Empirical Risk Minimization (ERM)

There is a vast literature on generalization bounds for empirical risk minimization (and natural variants) using tools such as (local) Rademacher complexity, stability, etc. (see e.g. [61, 41, 62] and many related references); however, many of these methods are not well-optimized for our setting because the noise and covariates are drawn from unbounded distributions and the squared loss is not uniformly Lipschitz (see the discussion in [63]). Fortunately, the framework developed in [63] avoids these issues and we are able to use it directly to give a good bound on the excess risk of the empirical risk minimizer.

### G.1.1  Background: Learning without Concentration Framework

We recall the main result of [63]. In this framework, as with many results in statistical learning, the empirical process is controlled by fixpoints of local Rademacher averages defined below. See e.g. [61] for more context.

Let $\mathcal{F}$ be a class of (measurable) functions. Let $X, Y$ be arbitrary random variables, suppose that $f^*$ is a minimizer of $\mathbb{E}[(Y - f(X))^2]$ over $f \in \mathcal{F}$ (which we assume exists) and define $\xi := Y - f^*(X)$. Let $\|f\|_{L_2} = \sqrt{\mathbb{E}[f^2]}$ and let $D_2(f)$ be the $L_2$ ball of radius 1 around $f$, i.e. $D_2(f) = \{g : \mathbb{E}[(g-f)^2] = 1\}$. The following two quantities, defined by fixed point equations, appear in the generalization bound: the intrinsic parameter (which does not depend on the noise model)

$$\beta_m^*(\gamma) = \inf\left\{r > 0 : \mathbb{E}\sup_{f \in \mathcal{F} \cap rD_{f^*}} \left|\frac{1}{\sqrt{m}}\sum_{i=1}^m \epsilon_i(f - f^*)(X_i)\right| \leq \gamma r \sqrt{m}\right\}$$

and the noise-sensitive parameter

$$\alpha_m^*(\gamma, \delta) = \inf\left\{s > 0 : \Pr\left(\sup_{f \in \mathcal{F} \cap sD_{f^*}} \left|\frac{1}{\sqrt{m}}\sum_{i=1}^m \epsilon_i\xi_i(f - f^*)(X_i)\right| \leq \gamma s^2 \sqrt{m}\right) \geq 1 - \delta\right\}.$$

**Theorem 13** (Theorem 3.1, [63]). *Suppose* $\mathcal{F}$ *is a closed, convex class of functions and* $f^*, X, Y, \alpha^*, \beta^*$ *are defined as above. Let* $\tau > 0$, *define*

$$q := \inf_{f \in \mathcal{F} - \mathcal{F}} \Pr(|f| \geq 2\tau \|f\|_{L_2})$$

*and assume that $q > 0$ (this is called the* small-ball *condition). Then for any $\gamma < \tau^2 q/16$ and for every $\delta > 0$ it holds that for any $\hat{f}$ which is an empirical risk minimizer for i.i.d. samples $\{(X^{(i)}, Y^{(i)})\}_{i=1}^m$,*

$$\|\hat{f} - f^*\|_{L_2} \leq 2 \max\{\alpha_m^*(\gamma, \delta/4), \beta_m^*(\tau q/16)\}$$

*with probability at least $1 - \delta - e^{-mq/2}$.*

### G.1.2 ERM Risk Bound

We return to the specific setting of hybrid $\ell_1$-constrained regression and prove our desired bound.

**Theorem 14.** *As long as $m = \Omega(\log(n/\delta))$, if $\hat{w}, \hat{a}'$ is the empirical risk minimizer for hybrid L1 regression from $m$ i.i.d. samples then*

$$\mathbb{E}[(\mathbb{E}[Y|X, Z] - \langle \hat{w}, X \rangle - \hat{a}'Z)^2] = O\left(RW\sigma\sqrt{\frac{\log(2n/\delta)}{m}} + \frac{\sigma^2 \log(4/\delta)}{m} + \frac{R^2 W^2 \log(n)}{m}\right)$$

*with probability at least $1 - \delta$.*

*Proof.* We first deal with the small-ball condition. Let $\tau = 1/2$. Observe that for any $f_1, f_2 \in \mathcal{F}$ that $f_1(X, Z) - f_2(X, Z)$ has a univariate Gaussian distribution, therefore

$$q := \Pr(|f| \geq 2\tau \|f\|_{L_2}) = 1 - \frac{1}{\sqrt{2\pi}} \int_{-2\tau}^{2\tau} e^{-x^2/2} dx \geq 1/4.$$

We take $\gamma = 1/300 < \tau^2 q/32$.

We now bound $\beta^*$. We have

$$\mathbb{E} \sup_{f \in \mathcal{F} \cap rD_{f^*}} \left| \frac{1}{\sqrt{m}} \sum_{i=1}^m \epsilon_i (f - f^*)(X_i) \right|$$

$$= \mathbb{E} \sup_{f \in \mathcal{F} \cap rD_{f^*}} \left| \frac{1}{\sqrt{m}} \sum_{i=1}^m \epsilon_i (\langle w - w^*, X_i - \mathbb{E}[X_i|Z_i]\rangle + (a - a^*)Z) \right|$$

$$\leq 2RW\mathbb{E} \left\| \frac{1}{\sqrt{m}} \sum_{i=1}^n \epsilon_i \frac{X_i - \mathbb{E}[X_i|Z_i]}{W} \right\|_\infty + \sup_{f \in \mathcal{F} \cap rD_{f^*}} |a - a^*| \mathbb{E}|Z|$$

$$\leq C(RW\sqrt{\log(n)} + \sup_{f \in \mathcal{F} \cap rD_{f^*}} |a - a^*|)$$

where the first inequality is by Holder's inequality and the triangle inequality, and the second is by the standard Gaussian tail bound combined with the union bound. To complete the bound observe that

$$\mathbb{E}[(\langle w - w^*, X - \mathbb{E}[X|Z]\rangle + (a - a^*)Z)^2] \geq (a - a^*)^2$$

so $a - a^* \leq r$ and

$$\mathbb{E} \sup_{f \in \mathcal{F} \cap rD_{f^*}} \left| \frac{1}{\sqrt{n}} \sum_{i=1}^m \epsilon_i (f - f^*)(X_i) \right| \leq C(RW\sqrt{\log(n)} + r).$$

This is smaller than $\gamma r\sqrt{m}$ as long as $r = \Omega(\frac{RW}{\gamma}\sqrt{\frac{\log n}{m}})$ so $\beta_m^* = O(\frac{RW}{\gamma}\sqrt{\frac{\log n}{m}})$.

We proceed to bound $\alpha^*$ similarly.

$$\sup_{f \in \mathcal{F} \cap sD_{f^*}} \left| \frac{1}{\sqrt{m}} \sum_{i=1}^m \epsilon_i \xi_i (f - f^*)(X_i) \right|$$

$$= \sup_{f \in \mathcal{F} \cap sD_{f^*}} \left| \frac{1}{\sqrt{m}} \sum_{i=1}^m \epsilon_i \xi_i (\langle w - w^*, X_i - \mathbb{E}[X_i|Z_i]\rangle + (a - a^*)Z) \right|$$

$$\leq C(RW\sigma\sqrt{\log(2n/\delta)} + \sigma s\sqrt{\log(4/\delta)})$$

with probability at least $1-\delta$ as long as $m \geq m_1 = O(\log(n/\delta))$, where the inequality is by Holder's inequality and $|a - a^*| \leq s$ (as before), Bernstein's inequality (Theorem 2.8.2 of [43]) using that the product of sub-Gaussian r.v. ($\xi_i$ and $X_i - \mathbb{E}[X_i|Z_i]$) is sub-exponential (Lemma 2.7.7 of [43]), and the union bound. The last quantity is upper bounded by $\gamma s^2 \sqrt{m}$ as long as $s^2 = \Omega(\frac{\sigma}{\gamma}\sqrt{\frac{\log(2n/\delta)}{m}})$ and $s = \Omega(\frac{\sigma}{\gamma}\sqrt{\frac{\log(4/\delta)}{m}})$. Therefore

$$(\alpha^*)^2 = O\left(\frac{RW\sigma}{\gamma}\sqrt{\frac{\log(2n/\delta)}{m}} + \frac{\sigma^2 \log(4/\delta)}{\gamma^2 m}\right).$$

Combining our estimates, it follows from Theorem 13 that

$$\mathbb{E}[(\hat{f} - f^*)^2] = O((\alpha_m^*)^2 + (\beta_m^*)^2) = O\left(\frac{RW\sigma}{\gamma}\sqrt{\frac{\log(2n/\delta)}{m}} + \frac{\sigma^2 \log(4/\delta)}{\gamma^2 m} + \frac{R^2 W^2 \log(n)}{\gamma m}\right)$$

with probability at least $1 - \delta - e^{-m/8} \geq 1 - 2\delta$ as long as $m = \Omega(\log(1/\delta) + m_1) = \Omega(\log(d/\delta))$. Since $\gamma$ is just a constant, this gives the result. $\qquad\square$

### G.2 Guarantees for Greedy Methods

In this section we show that a simple greedy method can also solve this high-dimensional regression problem with the correct dependence on $n$, albeit with slightly worse dependence on the other parameters. This is conceptually important as it shows that examples breaking greedy algorithms (in the sense of requiring $\omega(\log(n))$ sample complexity) also suffice to break analyses based on bounded $\ell_1$-norm.

**Lemma 23.** *In the hybrid $\ell_1$-regression model, there exists an input coordinate $j$ such that*

$$\mathrm{Var}(\mathbb{E}[Y|X, Z] \mid Z, X_j) \leq \mathrm{Var}(\mathbb{E}[Y|X, Z] \mid Z)\left(1 - \frac{\mathrm{Var}(\mathbb{E}[Y|X, Z] \mid Z)}{R^2 W^2}\right).$$

*Proof.* By expanding, applying Holder's inequality and using the assumption on $R$ we have

$$\mathrm{Var}(\mathbb{E}[Y|X, Z] \mid Z) = \sum_j w_j \mathrm{Cov}(\mathbb{E}[Y|X, Z], X_j \mid Z)$$
$$\leq W \max_j |\mathrm{Cov}(\mathbb{E}[Y|X, Z], X_j \mid Z)|$$
$$\leq RW \max_j \left|\mathrm{Cov}\left(\mathbb{E}[Y|X, Z], \frac{X_j}{\sqrt{\mathrm{Var}(X_j|Z)}} \mid Z\right)\right|.$$

Let $j$ be the maximizer. Then by Lemma 3,

$$\mathrm{Var}(\mathbb{E}[Y|X, Z] \mid Z) - \mathrm{Var}(\mathbb{E}[Y|X, Z] \mid Z, X_j) = \frac{\mathrm{Cov}(\mathbb{E}[Y|X, Z], X_j \mid Z)^2}{\mathrm{Var}(X_j \mid Z)} \geq \frac{\mathrm{Var}(\mathbb{E}[Y|X, Z] \mid Z)^2}{R^2 W^2}.$$

Rearranging gives that

$$\mathrm{Var}(\mathbb{E}[Y|X, Z] \mid Z, X_j) \leq \mathrm{Var}(\mathbb{E}[Y|X, Z] \mid Z)\left(1 - \frac{\mathrm{Var}(\mathbb{E}[Y|X, Z] \mid Z)}{R^2 W^2}\right).$$

$\qquad\square$

The above bound naturally leads to analyzing the recursion $x \mapsto x - cx^2$, which we do in the next Lemma.

**Lemma 24.** *Suppose that $x_1 \leq 1/2c$ and $x_{t+1} \leq (1 - cx_t)x_t$ for some $c < 1$. Then*

$$x_t \leq \frac{1}{c(t+1)}$$

*Proof.* We prove this by induction. Observe that $x(1 - cx)$ is an increasing function in $x$ for $x \leq \frac{1}{2c}$ since $1/2c$ corresponds to the vertex of the parabola, so using the assumption and the induction hypothesis,

$$x_t \leq x_{t-1}(1 - cx_{t-1}) \leq 1/ct - 1/ct^2 = \frac{t-1}{ct^2} \leq \frac{t-1}{c(t^2-1)} \leq \frac{1}{c(t+1)}.$$

$\square$

**Lemma 25.** *In the hybrid $\ell_1$-regression model,*

$$\mathrm{Var}(\mathbb{E}[Y|X,Z] \mid Z) \leq R^2 W^2.$$

*Proof.* By expanding, using Holder's inequality and Cauchy-Schwartz

$$\mathrm{Var}(\mathbb{E}[Y|X,Z] \mid Z) = \sum_j w_J \mathrm{Cov}(\mathbb{E}[Y|X,Z], X_j \mid Z)$$

$$\leq W \max_j |\mathrm{Cov}(\mathbb{E}[Y|X,Z], X_j \mid Z)|$$

$$\leq W \max_j \sqrt{\mathrm{Var}(\mathbb{E}[Y|X,Z] \mid Z)\mathrm{Var}(X_j \mid Z)} \leq RW \sqrt{\mathrm{Var}(\mathbb{E}[Y|X,Z] \mid Z)}$$

so $\mathrm{Var}(\mathbb{E}[Y|X,Z] \mid Z) \leq R^2 W^2$. $\square$

**Remark 6** (Connection to Approximate Caratheodory). *From the previous two lemmas, we can give a "matching pursuit" proof of the approximate Caratheodory theorem, which says that vectors of bounded $\ell_1$-norm are well approximated in $\ell_2$ by sparse vectors [43]. The standard proof of this result is probabilistic. Another proof, in a similar spirit, is given by using the guarantees of the Frank-Wolfe algorithm (see [64]).*

The remaining task is to analyze the behavior of the iteration under noise, which gives the main result:

**Theorem 15.** *For any $\epsilon \in (0,1)$, iterate $t$ of OMP in the hybrid regression model satisfies*

$$\mathrm{Var}(\mathbb{E}[Y|X,Z]|Z, X_{S_t}) \leq \epsilon\sigma^2$$

*as long as $t = \Omega(R^2 W^2 / \epsilon\sigma^2)$ and $m = \Omega(\frac{R^2 W^2}{\epsilon^2 \sigma^2}(t\log(4n) + \log(4/\delta)))$.*

*Proof.* The argument is structured similarly to the proof of Lemma 19. Fix $\epsilon \in (0,1)$ to be optimized later: we bound the number of steps of OMP during which $\mathrm{Var}(\mathbb{E}[Y|X,Z]|Z, X_{S_t}) \geq \epsilon\sigma^2$. Note that once this bounds holds for some $t$, it holds for all larger $t$ by the law of total variance. Fix an integer $T > 0$ to be optimized later.

First observe from Lemma 23 (applied after conditioning out $X_{S_t}$) that there exists a node $j^*$ such that

$$\mathrm{Var}(\mathbb{E}[Y|X,Z]|Z, X_{j^*}, X_{S_t}) \leq \mathrm{Var}(\mathbb{E}[Y|X,Z]|Z, X_{S_t})(1 - \frac{\mathrm{Var}(\mathbb{E}[Y|X,Z]|Z, X_{S_t})}{R^2 W^2}).$$

From Lemma 14 and taking the union bound over all sets $S$ of size $|S| \leq T$ we have

$$\left| \sqrt{\frac{1}{\mathrm{Var}(Y|X_{S\setminus j}) - \sigma^2}} \frac{|\hat{w}_j|}{\sqrt{(\hat{\Sigma}^{-1})_{jj}}} - \sqrt{\gamma'} \right| \leq \sqrt{\frac{\mathrm{Var}(Y|X_S)}{\mathrm{Var}(Y|X_{S\setminus j}) - \sigma^2} \cdot \frac{2\log(n^T/\delta)}{m}} + \sqrt{\frac{\gamma'}{64}} \leq \sqrt{\frac{1+\epsilon}{\epsilon} \cdot \frac{2\log(n^T/\delta)}{m}} + \sqrt{\frac{\gamma'}{64}}$$

using that $(1+x)/x = 1/x + 1$ is monotone decreasing, where

$$\gamma' = \gamma'(S, j) := \frac{\mathrm{Var}(X_i|Z, X_{S\setminus\{j\}}) - \mathrm{Var}(X_i|Z, X_S)}{\mathrm{Var}(X_i|Z, X_{S\setminus\{j\}}) - \sigma^2}.$$

Note that $\gamma'(S, j^*) \geq \epsilon\sigma^2/R^2 W^2$. Therefore as long as $m = \Omega(\frac{R^2 W^2}{\epsilon^2 \sigma^2}(T\log(4n) + \log(4/\delta)))$ then OMP chooses a node $j$ s.t.

$$\mathrm{Var}(\mathbb{E}[Y|X,Z]|Z, X_j, X_{S_t}) \leq \mathrm{Var}(\mathbb{E}[Y|X,Z]|Z, X_{S_t})(1 - \frac{\mathrm{Var}(\mathbb{E}[Y|X,Z]|Z, X_{S_t})}{2R^2 W^2})$$

as long as $|S_t| \leq T$. Applying Lemma 25 and Lemma 24 we find that

$$\text{Var}(\mathbb{E}[Y|X,Z]|Z,X_{S_t}) \leq \frac{2R^2W^2}{t+1}$$

for $t \leq T$. Therefore if $T \geq t \geq 2R^2W^2/\epsilon\sigma^2$ we are guaranteed that $\text{Var}(\mathbb{E}[Y|X,Z]|Z,X_{S_t}) \leq \epsilon\sigma^2$. Taking $\epsilon = 2R^2W^2/T\sigma^2$ gives the result. $\qquad\square$

## H   Regression and Structure Learning in Walk-Summable Models

### H.1   Failure of (weak) supermodularity in SDD models

The following example shows that the conditional variance is not supermodular in the SDD case, unlike in the attractive/ferromagnetic case.

**Example 5.** *Consider the GGM given by SDD precision matrix*

$$\Theta = \begin{bmatrix} 1 & -1/2 & -1/2 \\ -1/2 & 1 & 1/2 \\ -1/2 & 1/2 & 1 \end{bmatrix}$$

*and label the nodes (in order) by $i,j,k$. One can see (e.g. by computing effective resistances in the lifted graph) that $2\text{Var}(X_i) = 3$, that $2\text{Var}(X_i|X_j) = 2\text{Var}(X_i|X_k) = 8/3$, and $2\text{Var}(X_i|X_j,X_k) = 2$. Since $3 - 8/3 = 1/3 < 2/3 = 8/3 - 2$ this violates supermodularity.*

The above example alone does not rule out the possibility that (negative) conditional variances in SDD models always have *submodularity ratio* introduced by [32] lower bounded by a constant. We recall the definition next:

**Definition 16** ([32]). *The* submodularity ratio $\gamma(k)$ *of a function on subsets of a universe $U$, $f : 2^U \to \mathbb{R}_{\geq 0}$ is defined to be*

$$\gamma(k) := \min_{L \subset U, |S| \leq k, L \cap S = \emptyset} \frac{\sum_{x \in S} f(L \cup \{x\}) - f(L)}{f(L \cup S) - f(L)}$$

*Note that $\gamma(k) \geq 1$ for a submodular function.*

The significance of this ratio for a function $f$ is that if the ratio is lower bounded by a constant then similar guarantees for submodular maximization follow ([32]); for this reason such an $f$ is sometimes called *weakly submodular* (as in e.g. [65]). Now, we give a counterexample showing that for general SDD matrices, this ratio can be arbitrarily small.

**Example 6.** *Fix $M > 0$ large. Let $\epsilon > 0$ be a parameter to be taken small, and consider the following precision matrix, which is SDD as long as $\epsilon < 1/2 < M$:*

$$\Theta = \begin{bmatrix} 1 & -\epsilon & \epsilon \\ -\epsilon & M & \epsilon - M \\ \epsilon & \epsilon - M & M \end{bmatrix}.$$

*This has inverse*

$$\Theta^{-1} = \begin{bmatrix} (\epsilon - 2M)/(\epsilon + 2\epsilon^2 - 2M) & -(\epsilon/(\epsilon + 2\epsilon^2 - 2M)) & \epsilon/(\epsilon + 2\epsilon^2 - 2M) \\ -(\epsilon/(\epsilon + 2\epsilon^2 - 2M)) & (\epsilon^2 - M)/(\epsilon^2 + 2\epsilon^3 - 2\epsilon M) & (\epsilon + \epsilon^2 - M)/(\epsilon^2 + 2\epsilon^3 - 2\epsilon M) \\ \epsilon/(\epsilon + 2\epsilon^2 - 2M) & (\epsilon + \epsilon^2 - M)/(\epsilon^2 + 2\epsilon^3 - 2\epsilon M) & (\epsilon^2 - M)/(\epsilon^2 + 2\epsilon^3 - 2\epsilon M) \end{bmatrix}$$

*so*

$$\text{Var}(X_1) - \frac{1}{\Theta_{11}} = \frac{-2\epsilon^2}{\epsilon + 2\epsilon^2 - 2M}$$

*and (by computing the inverse of the top-left 2x2 submatrix of $\Theta$) we find*

$$\text{Var}(X_1|X_3) - \frac{1}{\Theta_{11}} = \frac{M}{M - \epsilon^2} - 1 = \frac{\epsilon^2}{M - \epsilon^2}$$

*and the difference is*

$$\text{Var}(X_1) - \text{Var}(X_3) = \frac{\epsilon^3}{(M - \epsilon^2)(2M - 2\epsilon^2 - \epsilon)}$$

Algorithm WS-REGRESSION($\gamma, d$):

1. Choose $j$ to minimize $\widehat{\text{Var}}(X_i | X_j)$.

2. Let $s_0^2 := \exp(\lfloor \log(\widehat{\text{Var}}(X_i | X_j)/8d) \rfloor - 1)$.

3. For $\ell$ in 0 to $\lceil \log(8d) + 3 \rceil$:

    (a) Let $s_\ell^2 := s_0 e^\ell$

    (b) Solve for $w, a$ in

    $$\min_{w, a: \|w\|_1 \leq \lambda} \hat{\mathbb{E}}_2 \left[ \left( X_i - \sum_{k \notin \{i,j\}} w_k \frac{X_k}{\sqrt{\widehat{\text{Var}}(X_k | X_j)}} - a X_j \right)^2 \right]$$

    where $\lambda = \sqrt{2d} s_\ell$ and $\hat{\mathbb{E}}_2$ is empirical expectation over sample set 2.

    (c) Let $\hat{\sigma}^2 := \hat{\mathbb{E}}_3 \left[ \left( X_i - \sum_{k \notin \{i,j\}} w_k \frac{X_k}{\sqrt{\widehat{\text{Var}}(X_k | X_j)}} - a X_j \right)^2 \right]$ where $\hat{\mathbb{E}}_3$ is empirical expectation over sample set 3. If $\lambda^2 \geq 2d\gamma^2 \hat{\sigma}^2$ (equivalently, $s_\ell^2 \geq \gamma^2 \hat{\sigma}^2$), then exit the loop.

4. Return $w, a, j, \hat{\sigma}^2$.

*Therefore the* submodularity ratio $\gamma = \gamma(2)$ *for* $f(S) = \text{Var}(X_1) - \text{Var}(X_1 | X_S)$ *is upper bounded by (taking* $L = \emptyset$*)*

$$\gamma \leq \frac{f(\{2\}) + f(\{3\})}{f(\{2,3\})} = \Theta \left( \frac{\epsilon^3/M^2}{\epsilon^2/M} \right) = \Theta(\epsilon/M)$$

*which is clearly arbitrarily small.*

**Remark 7** (Submodularity ratio and $\kappa$). *It's possible to show, based on Lemma 23 and the bounds in the proof of Theorem 16 to derive a partial lower bound for the submodularity ratio when we consider $S \subset T$ and restrict to $j$ which are neighbors of $i$, by showing:*

$$f(S \cup \{j\}) - f(S) \geq \frac{\kappa^2}{4d}(f(U) - f(S)) \geq \frac{\kappa^2}{4d}(f(T \cup \{j\}) - f(T))$$

*using the monotonicity of $f$ (which follows from the law of total variance) in the last step, and under the assumption that the model is $\kappa$-nondegenerate and $d$-sparse. The above example shows that this dependence on $\kappa$ is tight: by taking a fixed small $\epsilon$ and sending $M \to \infty$, the submodularity ratio can be as small as $O(\kappa^2)$ since $\kappa = \epsilon/\sqrt{M}$ in this model. It remains unclear if the submodularity ratio can be lower bounded in general in $\kappa$-nondegenerate models; even if such a bound did hold it could not be used to prove Theorem 16 since that result holds without a $\kappa$-nondegeneracy assumption.*

### H.2 Sparse regression

In this section we describe an algorithm to find a good predictor of node $X_i$ with bounded degree $d$ in a walk-summable GGM. To simplify the analysis, we assume the data has been split into 3 equally sized sample sets, each of size $m$; when there is no explicit mention, averages are taken over sample set 1.

The algorithm is conceptually straightforward: it does a single greedy step and then sets up an $\ell_1$-constrained regression. The only complication is that we do not know $1/\Theta_{ii}$ a priori, but this appears in the $\ell_1$-norm of the obvious regression we want to setup. Since we have multiplicative estimates for $1/\Theta_{ii}$, we can deal with this by searching over the possible values on a log scale.

We show this algorithm gives a result for sparse linear regression under the walk-summability assumption which (1) depends on sparsity only, not on norms (unlike the slow rate bound for LASSO) and (2) is computationally efficient (unlike brute force $\ell_0$-constrained regression).

**Theorem 16.** *Let $i$ be a node of degree $d$ in an SDD GGM and $\sigma^2 := 1/\Theta_{ii}$. Then WS-Regression($\gamma$) with $\gamma^2 = 2$ returns $w, a$ such that*

$$\mathbb{E}\left[\left(\mathbb{E}[X_i|X_{\sim i}] - \sum_{k \notin \{i,j\}} w_k \frac{X_k}{\sqrt{\widehat{\text{Var}}(X_k|X_j)}} - aX_j\right)^2\right] = O\left(\sigma^2 \sqrt{\frac{d\log(2n/\delta)}{m}}\right)$$

*and $\hat{\sigma}^2$ s.t. $1/2 \leq \Theta_{ii}\hat{\sigma}^2 \leq 2$ with probability at least $1 - \delta$, as long as $m = \Omega(\log(n/\delta))$.*

*Proof.* By Lemma 8, for any $k \sim i$ we have $\text{Var}(X_i|X_j) \leq 1/|\Theta_{ik}|$ therefore if we take $j^*$ which minimizes $\text{Var}(X_i|X_{j^*})$ then
$$\text{Var}(X_i|X_{j^*}) \leq 1/|\Theta_{ij}|$$
for all $j$. Similarly, applying Lemma 9 we know that
$$\text{Var}(X_i|X_{j^*}) \leq \frac{4d}{\Theta_{ii}}$$

By using Lemma 10 and taking the union bound over the randomness of sample set 1, we may assume that for every $j, k$, $\text{Var}(X_k|X_j)/\sqrt{2} \leq \widehat{\text{Var}}(X_k|X_j) \leq \sqrt{2}\text{Var}(X_k|X_j)$, with probability at least $1 - \delta/3$ as long as $m = \Omega(\log(n/\delta))$. We condition on this event. Then for the $j$ chosen in step 1 of the algorithm, we have that

$$\text{Var}(X_i|X_j) \leq \sqrt{2}\widehat{\text{Var}}(X_i|X_j) \leq \sqrt{2}\widehat{\text{Var}}(X_i|X_{j^*}) \leq 2\text{Var}(X_i|X_{j^*}) \leq 2/|\Theta_{ik}|$$

for all $i \sim k$, and similarly

$$\text{Var}(X_i|X_j) \leq \frac{8d}{\Theta_{ii}}. \tag{10}$$

Furthermore,

$$\text{Var}\left(\frac{X_k}{\sqrt{\widehat{\text{Var}}(X_k|X_j)}} \middle| X_j\right) \leq \sqrt{2}$$

and

$$\begin{aligned}
\sum_k \frac{|\Theta_{ik}|}{\Theta_{ii}}\sqrt{\widehat{\text{Var}}(X_k|X_j)} &\leq \sum_k \frac{|\Theta_{ik}|}{\Theta_{ii}}\sqrt{2\text{Var}(X_k|X_j)} \\
&\leq \sum_k \frac{|\Theta_{ik}|}{\Theta_{ii}}\sqrt{2(1/|\Theta_{ik}| + \text{Var}(X_i|X_j))} \\
&\leq \sum_k \frac{|\Theta_{ik}|}{\Theta_{ii}}\sqrt{2(3/|\Theta_{ik}|)} = \frac{\sqrt{6}}{\Theta_{ii}}\sum_k \sqrt{|\Theta_{ik}|} \leq \sqrt{6d/\Theta_{ii}}
\end{aligned}$$

using Lemma 7 in the second inequality and Cauchy-Schwartz and the SDD property in the final inequality. Given (10) we know that for one of the values of $\ell$ satisfies $e/\Theta_{ii} \leq s_\ell^2 \leq e^2/\Theta_{ii}$; call this $\ell^*$. By Theorem 14 we have that with probability at least $1-\delta/3$ that for all of the loop iterations where $1/\Theta_{ii} \leq s_\ell^2$ (so the global optimal $w^*, a$ is in the constraint set) and $\ell \leq \ell^*$

$$\mathbb{E}\left[\left(X_i - \sum_{k \notin \{i,j\}} w_k \frac{X_k}{\sqrt{\text{Var}(X_k|X_j)}} - aX_j\right)^2\right] = O\left(\sqrt{1/\Theta_{ii}}\sqrt{24d/\Theta_{ii}}\sqrt{2}\sqrt{\frac{\log(n^2/\delta)}{m}}\right) \tag{11}$$

as long as $m = \Omega(\log(n/\delta))$, using that $d \leq n$ in the union bound. Condition on this and consider only the randomness over sample set 3. By Bernstein's inequality and the union bound over the loop iterations, with probability at least $1 - \delta/3$ as long as $m = \Omega(\log(n/\delta))$, for the above value of $\ell = \ell^*$ we have that the test in 3(c) succeeds and the loop exits, and that if the loop exited in a previous iteration then $\frac{1}{\Theta_{ii}} = \text{Var}(X_i|X_{\sim i}) \leq s_\ell^2$ so we can apply the above guarantee (11), giving the result. $\qquad\square$

Algorithm HYBRIDMB$(\tau, \gamma, d)$:

1. We suppose the samples are split into 3 equally sized sets as in the description of WS-REGRESSION.

2. For every node $i$, apply WS-REGRESSION which returns $w(i), a(i), j(i), \hat{\sigma}^2(i)$.

3. Define $u(i)_{j(i)} = a(i)$ and $u(i)_k = \frac{w(i)_k}{\sqrt{\widehat{\mathrm{Var}}(X_k|X_j)}}$.

4. Let $E = \{\}$.

5. For every pair of nodes $a, b$:

   (a) If $u(a)_b^2 \hat{\sigma}^2(b) \geq \tau \hat{\sigma}^2(a)$ and $u(b)_a^2 \hat{\sigma}^2(a) \geq \tau \hat{\sigma}^2(b)$: add $(i, j)$ to $E$.

6. Return edge set $E$.

## H.3   Structure learning

**Theorem 17.** *For an SDD, $\kappa$-nondegenerate GGM the following is true. Algorithm HYBRIDMB with $\tau = \kappa^2/8, \gamma = 2$ returns the true neighborhood of every node $i$ with probability at least $1 - \delta$ as long as $m \geq m_1'$, where $m_1' = O((d/\kappa^4) \log(n/\delta))$ where $d$ is the max degree in the graph.*

*Proof.* By Theorem 16 and the union bound, we may assume with probability at least $1 - \delta$, as long as $m = \Omega((d/\kappa^4) \log(n/\delta))$ that for every node $i$ we have $u(i)$ such that

$$\mathbb{E}\left[ \left( \mathbb{E}[X_i|X_{\sim i}] - \sum_{k \neq i} u(k) X_k \right)^2 \right] \leq \frac{\kappa^2}{16 \Theta_{ii}}$$

and $\hat{\sigma}^2(i)$ which is within a factor of 2 of $1/\Theta_{ii}$. Applying the law of total variance and (**??**) we find that

$$\left( \frac{u(k)}{\sqrt{\Theta_{kk}}} + \frac{\Theta_{ik}}{\Theta_{ii}\sqrt{\Theta_{kk}}} \right)^2 = \left( u(k) + \frac{\Theta_{ik}}{\Theta_{ii}} \right)^2 \mathrm{Var}(X_k|X_{\sim k}) \leq \frac{\kappa^2}{64 \Theta_{ii}}$$

so if $i$ and $k$ are not neighbors, then $\Theta_{ik} = 0$ so

$$u(k)^2 \hat{\sigma}^2(k) \leq 2u(k)^2/\Theta_{kk} \leq \frac{\kappa^2 \hat{\sigma}_i^2}{16}$$

and if they are then $|\Theta_{ik}| \geq \kappa \sqrt{\Theta_{ii}\Theta_{kk}}$ so using the reverse triangle inequality

$$u(k)^2 \hat{\sigma}^2(k) \geq (1/2)u(k)^2/\Theta_{kk} \geq (1/2)(\kappa^2/\sqrt{\Theta_{ii}} - \kappa/8\sqrt{\Theta_{ii}}) \geq (7/16)\kappa^2/\sqrt{\Theta_{ii}} \geq (7/32)\kappa^2 \hat{\sigma}^2(i).$$

From these inequalities we see that in step 5 (a) exactly the correct edges are chosen.   $\square$

**Theorem 18.** *For any SDD, $\kappa$-nondegenerate GGM the following is true. Algorithm GREEDYAND-PRUNE with $\tau = \kappa^2/8$ and $T = \Theta(d/\kappa^2)$ returns the true neighborhood of every node $i$ with probability at least $1 - \delta$ as long as $m = \Omega((d^2/\kappa^6) \log(n/\delta))$ where $d$ is the max degree in the graph.*

*Proof.* The proof is the same as for Theorem 17 except that we use Theorem 15 instead of Theorem 14, and use the slightly different pruning analysis from the proof of Theorem 7.   $\square$

**Remark 8** (Implementation)**.** *In experiments, to reduce the number of free parameters in HYBRIDMB we define $\gamma' = 2d\gamma^2$ and note that using $\gamma'$ instead of $\gamma$ actually allows $d$ to be eliminated as a parameter. We also use a single sample set instead of sample splitting; we expect that the algorithm can still be proved correct without the splitting, at the cost of a more lengthy analysis.*

For completeness, we state a result for HYBRIDMB under the $\ell_1$-bounded assumption used in previous work like [58, 42]. The proof follows the proof of our main result, except that we can ignore the analysis of the first greedy step and simply use the a priori estimate for the $\ell_1$ norm, which only shrinks under conditioning.

**Theorem 19.** *For any $\kappa$-nondegenerate GGM with precision matrix $\Theta : n \times n$ such that $\max_i \sum_{j=1}^n |\Theta_{ij}| \le M$, Algorithm HYBRIDMB with $\tau = O(\kappa^2)$ returns the true neighborhood of every node $i$ with probability at least $1 - \delta$ as long as $m = \Omega(M^2 \log(n/\delta)/\kappa^4)$.*

This guarantee matches [42], which itself improves on the guarantee in [58]. In the same setting, GREEDYANDPRUNE achieves a sample complexity of $O(\frac{M^4 \log(n/\delta)}{\kappa^6})$.

# I Simulations and Experiments

In this section, we will compare our proposed method (GREEDYANDPRUNE) with popular methods previously introduced in the literature: the Graphical Lasso [9], the Meinhausen-Bühlmann estimator (based on the LASSO) [10], CLIME [58], and ACLIME [42] (an adaptive version of CLIME). In the first subsection, we consider simple attractive GGMS and show that our method always performs well compared to previous methods and sometimes outperforms them considerably. In the second subsection, we compare the performance on a real dataset (from [50]) and show that our methods HYBRIDMB and GREEDYANDPRUNE again compare favorably. Our experiment also gives evidence that walk-summability is a reasonable assumption in practice.

## I.1 Simple attractive GGMs where previous methods perform poorly

Three of the most popular methods for recovering a sparse precision matrix in practice are the Graphical Lasso (glasso) [9], the Meinhausen-Bühlmann estimator (MB) based on the Lasso [10], and the CLIME estimator [58]. The graphical lasso is the $\ell_1$-penalized variant of the MLE (Maximum Likelihood Estimator) for the covariance matrix; CLIME minimizes the $\ell_1$-norm of the recovered precision matrix $\hat{\Theta}$, given an $\ell_\infty$ constraint $|\Sigma\Omega - Id|_\infty \le \lambda$ (where $|M|_\infty = \|M\|_{1\to\infty}$ is the entrywise max-norm). For Meinhausen-Bühlmann, we let the estimated $\hat{\Theta}$ have its rows be given by the appropriate lasso estimate, scaled appropriately by the corresponding estimate for the conditional variance. The current theoretical guarantees of these methods have very high sample complexity for general GFFs and we find simple examples in which the scaling of their sample complexity with $n$ is poor. One example (which breaks the Meinhausen-Bühlmann estimator) is simply based off of a simple random walk observed at large times; the other examples we use are simple combinations of a path and cliques:

**Example 7** (Path and cliques). *Fix $d$ and suppose $n/2$ is a multiple of $d$. Let $B$ be a standard Brownian motion in 1 dimension, and let $X_1, \ldots, X_{n/2}$ be the values of the $B$ at equally spaced points in the interval $[1/2, 3/2]$, i.e. $X_1 = B(1/2), X_2 = B((1/2) + 1/(n-1)), \ldots$ Equivalently, let the covariance matrix of this block be $\mathrm{Cov}(X_i, X_j) = 1/2 + \min(i,j)/n$, or take the Laplacian of the path and add the appropriate constant to the top-left entry.*

*Let the variables $X_{n/2+1}, \ldots, X_n$ be independent of the Brownian motion, and let their precision matrix be block-diagonal with $d \times d$ blocks of the form $\Theta_1$ where $\Theta_1$ is a rescaling of $\Theta_0$ so that the coordinates have unit variance, and $\Theta_0 = I - (\rho/d)\vec{1}\vec{1}^T$ where $\rho \in (0,1)$. In all experiments, we finally standardize the variables to have unit variance, following the usual recommendation (although the variances in this example are already bounded between $0.5$ and $1.5$).*

The results of running all methods[11] on samples from this model are shown in Figure 2 for the Frobenius error with a fixed number of samples ($m = 150$) where the clique degree is $d = 4$ and the edge strength is $\rho = 0.95$. In Figure 3 we show the number of samples needed to recover the true edge structure for the same example with $d = 4$ in two cases, $\rho = 0.7$ and $\rho = 0.95$. We note that our definition of structure recovery is fairly generous — we apply a thresholding operation to the returned $\Theta$ matrix using the true value of $\kappa/2$, so the algorithms are not penalized for returning matrices with many small nonzero entries (which happens in practice at the optimal tuning of parameters, even though in the theory of e.g. [10] neighborhood estimates are made just from the support of the lasso estimate).

Figure 2: Normalized error (measured by $\|\hat{\Theta} - \Theta\|_1/n$ where $\|\cdot\|_1$ denotes the $\ell_1$ norm viewing the matrix as a vector) in the precision matrix returned in Example 7 with $\rho = 0.95$. We note that this quantity should be expected to scale at least linearly, because some entries of $\Theta$ grow with $n$. Errors were averaged over 8 trials for each $n$ and hyperparameters were chosen by grid search minimizing the recovery error in a separate trial, for each value of $n$. The tested parameters for $\lambda$ in glasso were chosen from a log grid with 15 points from $0.0005$ to $0.4$, similarly for $\lambda$ in MB, from 8 points from 1 to 32 for $\gamma'$ in HYBRIDMB (we set $\tau = 0$ for a more direct comparison to MB), for CLIME from a log grid with 15 points from $0.01$ to $0.8$, and for GREEDYANDPRUNE $k$ from a rounded log grid with 7 points from 3 to 24 and $\nu$ from a log grid with 8 points from $0.001$ to $0.1$.

Note in particular that from Figure 3, we see the sample complexity of GREEDYANDPRUNE scales like $O(\log(n))$, the information-theoretic optimal scaling which is in agreement with Theorem 7, while in the first example ($\rho = 0.7$) the sample complexity of the Graphical Lasso scales roughly like $\Theta(n)$ and in the second example ($\rho = 0.95$) the same is true for CLIME.

Recall that these examples are well-outside of the regime where the theoretical guarantees for methods like CLIME and Graphical Lasso can guarantee accurate reconstruction from $O(polylog(n))$ sammples, which is one reason we might expect them to be hard in practice. For example, the analysis of CLIME requires a bound on the entries of the inverse covariance (after rescaling the coordinates to have variance $\Theta(1)$), but for the path Laplacian the entries of the precision matrix are of order $\Theta(n)$.

We describe one additional intuition as to why the Graphical Lasso should fails on this example: for the penalty $\lambda\|\hat{\Theta}\|_1$ to respect the structure of the path (where conditional variances are small) $\lambda$ should be chosen small, but then the nodes in the cliques may gain spurious edges to the path and other cliques. With CLIME there is a similar concern that the $\ell_1$ penalty for the two types of nodes does not scale properly. Different regularization parameters for the different types of edges could help in this particular example — however, it is typically difficult know beforehand which nodes have small and big conditional variances without effectively learning the GGM, as the way to show a node has low conditional variance almost always involves finding a good predictor of it from the other nodes. Concretely, in the case of ACLIME, it performed significantly worse than CLIME in most of our tests. On the other hand, the rescaling performed by our proposed algorithm HYBRIDMB does resolve this issue in a principled way.

In the above two examples we tried, the (thresholded) Meinhausen-Bühlmann estimator successfully achieved similar sample complexity to our proposed methods, despite the fact that this example is again well outside of the regime where its theoretical guarantees are good. However, as we see in Figure 4 the sample complexity of this estimator is poor in another very simple example: a simple random walk with Gaussian steps run from times $n$ to $2n$. (As before, this is the description of the model before standardizing coordinates to variance 1.) This is again not so surprising, as we know the Lasso (which the MB method is based upon) can only be guaranteed to obtain its "slow rate" guarantee when the coordinates of the input are highly dependent, and the slow rate guarantee for Lasso depends on norm parameters that are not sufficiently small in our example for good recovery guarantee.

Figure 3: (a) $d = 4$ and $\rho = 0.7$, (b) $d = 4$ and $\rho = 0.95$. Number of samples needed to approximately recover true edge structure after thresholding using the test $\frac{|\hat{\Theta}_{ij}|}{\sqrt{\hat{\Theta}_{ii}\hat{\Theta}_{jj}}} > \kappa/2$, where $\kappa$ is the $\kappa$ for the true precision matrix from the information-theoretic assumption (1). Samples are drawn from the model in Example 7 with two different values for the edge strength $\rho$. Note that the sample complexity of GREEDYANDPRUNE is consistent with the $O(\log(n))$ bound established in Theorem 7, whereas the graphical lasso and CLIME have sample complexity that appears to be roughly $\Theta(n)$ in the left and right examples respectively. The $m$ shown is the minimal number of samples needed for the average number of incorrect edges per node (counting both insertions and deletions) to be at most 1. Trials and parameter selection was performed the same way as in the experiment for Figure 2, except that the parameters were chosen to minimize the number of incorrect edges, instead of error in the $\ell_1$ norm.

Figure 4: Large initial time simple random walk example: the setup is the same as in Figure 3, except that the ground truth model is a Gaussian simple random walk observed from times $n$ to $2n$. We observe in this example that the sample complexity of ACLIME and the Lasso-based Meinhausen-Bühlmann estimator appear to scale roughly linearly in $n$, whereas the sample complexity of GREEDYANDPRUNE and HYBRIDMB is in fact constant over the observed values of $n$.

| Method | CV Error | CV Parameters | # Non-zeros | Cond. No. | $M$ | $\Delta_{WS}$ |
|---|---|---|---|---|---|---|
| Graphical Lasso | 0.13 | $\lambda = 0.01$ | 4378 | 968.6 | 54.8 | 8.7 % |
| CLIME | 0.41 | $\lambda = 0.21$ | 806 | 193.8 | 232.2 | 0.0 % |
| GREEDYANDPRUNE | 0.27 | $k = 13, \nu = 0.01$ | 476 | 389.4 | 224 | 1.1 % |
| MB | 0.17 | $\lambda = 0.05$ | 1854 | 21439 | 156 | 1.1 % |
| HYBRIDMB | 0.19 | $\gamma' = 21$ | 2758 | 1080843 | 324 | 2.2 % |

Table 1: Results for precision matrix selected via 5-fold CV on Riboflavin dataset. The last 4 columns give summary statistics for the final recovered $\hat{\Theta}$ using the CV parameters on the entire dataset: $M$ is the maximum $\ell_1$ row norm for any row of $\Theta$, the same as in the guarantee for CLIME cited earlier. The walk-summable relative error is $\Delta_{WS} := \frac{\|\tilde{\Theta} - \hat{\Theta}\|_F}{\|\hat{\Theta}\|_F}$ where $\tilde{\Theta}$ is the closest walk-summable matrix to $\hat{\Theta}$ in Frobenius norm. This shows that all of the estimated precision matrices are either walk-summable or close to walk-summable.

| Method | Runtime (seconds) |
|---|---|
| Graphical Lasso | 0.74 |
| CLIME | 2.12 |
| GREEDYANDPRUNE | 0.19 |
| MB | 0.48 |
| HYBRIDMB | 1.84 |

Table 2: Sequential runtime of methods on Riboflavin dataset with CV parameters, averaged over 10 runs. In all experiments, the graphical lasso implementation was from the glasso R package, CLIME was implemented by calling Gurobi from R (due to numerical limitations of the standard package), MB and HYBRIDMB were implemented using the glmnet package, and for GREEDYANDPRUNE we used a naive R implementation.

## I.2    Results for Riboflavin dataset

In this section we analyze the behavior of recovery algorithms on a popular dataset provided in [50]. This dataset has $m = 71$ samples and describes (log) expression levels for $n = 100$ genes in *B. subtilis*. We compared all of the methods listed above; our tables do not list the ACLIME results because it did not achieve nontrivial reconstruction (it's CV error as defined below was 0.98, which is essentially the same as the score for returning the identity matrix). We selected parameters using a 5-fold crossvalidation with the following least-squares style crossvalidation objective[12], after standardizing the coordinates to each have empirical variance 1 and mean 0:

$$E(\hat{\Theta}) := \frac{1}{nm_{holdout}} \sum_{i=1}^{n} \sum_{k=1}^{m_{holdout}} (X_i^{(k)} + \sum_{j \neq i} \frac{\hat{\Theta}_{ij} + \hat{\Theta}_{ji}}{2\hat{\Theta}_{ii}} X_i^{(k)})^2.$$

Note that the true $\Theta$ minimizes this objective as $m_{holdout} \to \infty$, making it equal to the sum of conditional variances; when the initial variances are set to 1, this objective simply measures the average amount of variance reduction achieved over the coordinates.

The results of the cross-validation process[13] are shown in Table 1. As we see from the first 2 columns of the table, Graphical Lasso achieved the greatest amount of variance reduction but returned the densest estimate for $\Theta$, MB and HYBRIDMB had slightly less variance reduction, GREEDYAND-PRUNE had the sparsest estimate and achieved significantly more variance reduction that CLIME. We see that the chosen precision matrices have large condition number and row $\ell_1$-norm $M$, comparable to the number of nodes $n$, which is significant in that known guarantees for Graphical Lasso, MB, CLIME and ACLIME are only interesting when these quantities are small (e.g. constant or $O(\log n)$). (Equivalently, the gap between variance and conditional variance is large; we note that the true gap may be even larger if we had access to more data, since we might be able to find even

| Method | Number of Samples Needed | Optimal Parameters |
|---|---|---|
| Graphical Lasso | 500 | $\lambda = 0.005$ |
| CLIME | 550 | $\lambda = 0.04$ |
| GREEDYANDPRUNE | 550 | $k = 6, \nu = 0.01$ |
| MB | 550 | $\lambda = 0.01$ |
| HYBRIDMB | 525 | $\gamma' = 21$ |

Table 3: Number of samples needed to achieve error of at most $0.25$ incorrect edges per node after thresholding in the semi-synthetic experiment: samples were drawn from a $\Theta$ given by thresholding the graphical lasso estimate from the Riboflavin dataset. The details of the thresholding, etc. are the same as in the synthetic experiment of Figure 3.

Figure 5: Left: thresholded graph from graphical lasso output on riboflavin data, used in semisynthetic experiment (see Table 3). Right: unthresholded graph output by GREEDYANDPRUNE on Riboflavin data.

better estimators for each $X_i$ given the other coordinates.) On the other hand, the recovered matrices are not far from walk-summable in Frobenius norm, suggesting that this is indeed a reasonable assumption.

In Table 2 we record the sequential runtimes of all of the methods on this dataset using the CV parameters. GREEDYANDPRUNE was the fastest method. For larger datasets it is important to use parallelism, and we note we note that CLIME, MB, HYBRID.MB and GREEDYANDPRUNE are "embarassingly parallelizable", as each node can be solved independently, but this is not the case for the Graphical Lasso. In practice, on our synthetic datasets and using 24 cores, CLIME becomes faster than the Graphical Lasso and GREEDYANDPRUNE stays the fastest. In our experiment, we did not test our proposed method SEARCHANDVALIDATE or the methods of [14], although they have good sample complexity guarantees, due to computational limitations; in [14], they report their methods requires on the order of days to run on this example.

We also performed a "semi-synthetic" experiment on this dataset, by taking the recovered (dense) $\Theta$ from Graphical Lasso, thresholding it to have $\kappa = 0.15$ and computing the sample complexity to recover the edges of the graphical model from sampled data (as in the synthetic experiments, with error of at most $0.25$ incorrect edges per node, after thresholding at $\kappa/2$). All methods performed similarly on this test: the results are shown in Table 3.

**Remark 9.** *Several papers have been written on faster implementations of the graphical lasso, e.g. the Big & Quic estimator of [66]. However, these methods have mostly been developed/tested in the regime where $\lambda$ is quite large: e.g. the documentation for the R package BigQuic implementing Big & Quic suggests using $\lambda \geq 0.4$ and that $\lambda = 0.1$ is too small to run in a reasonable time on large datasets. In practice, these methods may even fail to return the true optimum when given small $\lambda$; however, the above experiment suggests this is an important regime in practice.*

# J   Some difficult examples

In this appendix, we continue the discussion of difficult non-walk summable examples from Section 4.

**Remark 10.** *Part of the motivation for the use of nearly-duplicated random variables is that one can prove (using essentially a modified version of Lemma 23)) that in a general sparse GGM there always exists at least one node $i$ with at least one neighbor $j$ such that $\mathrm{Var}(X_i|X_j)$ is noticeably smaller than $\mathrm{Var}(X_i)$. In this example, this is trivially true but is not useful for discovering connections between unpaired variables.*

**Example 8** (Harder Example). *The previous example, while it breaks* GREEDYANDPRUNE, *cannot be a hard example in general because the edge structure is easy to determine from the covariance matrix. (The covariance matrix is roughly block diagonal and each block corresponds to a clique). The following variant seems significantly harder: start with $\Sigma_0$ from the previous example, and then Schur complement (i.e. condition) out $d/4$ many of the nodes to yield $\Sigma_0'$. Then the covariance matrix of the whole model is block diagonal with $\Sigma_0'$ repeated $n/(d/4)$ times. Finally, we randomly permute the rows/columns.*

Experimentally, it seems that Example 8 breaks the methods considered in our experiments in the high-dimensional regime where the number of samples is much less than the dimension $n$. However, this example itself cannot be computationally hard to learn: a simple algorithm to learn it thresholds the covariance matrix to find the sub-blocks made up of the paired nodes from a block, then picks a sub-block, conditions it out, and finds the remaining nodes from this block as the nodes whose conditional variance went down significantly.

## Footnotes

[10]In an alternate convention which we do not use, $\mathrm{Var}(X|Y)$ is defined to be the random variable $\mathbb{E}[(X - \mathbb{E}[X|Y])^2|Y]$ and our definition is the same as $\mathbb{E}\mathrm{Var}(X|Y)$.

[11]For the Graphical Lasso we used the standard R packages recommended in the original papers. For CLIME, we originally tested the standard R package but it was unable to reconstruct a path, presumably due to numerical issues. To fix this, we reimplemented CLIME using Gurobi and used a similar implementation for ACLIME.

[12]An alternative which is sometimes used is the likelihood objective $\mathrm{Tr}(\hat{\Sigma}\hat{\Theta}) - \log\det(\hat{\Theta})$, but this objective is not very smooth due to the $\log\det$ term and may equal $\infty$ even for entry-wise "good" reconstructions.

[13]Essentially the same as before, parameters for Graphical Lasso were chosen from a log-scale grid from 0.001 to 0.5 with 15 points, for CLIME similarly from 0.01 to 0.8 with 20 points, and for GREEDYANDPRUNE from a rounded log-scale grid from 3 to 26 with 7 points and from 0.001 to 0.1 with 8 points.