[Reviews · NeurIPS 2020]

Review 1

Summary and Contributions: The paper focuses on the problem of structure learning in Gaussian graphical models with additional constraints like attractiveness or walk-summability. It is shown that these special subclasses indeed enable efficient (polynomial-time) algorithms with superior sample complexity bounds. An important contribution of the paper is also providing some better intuition for which are the challenging examples for GGM structure recovery.

Strengths: The paper seems fairly well written and the math is convincing. I agree that the bounds on sample complexity should be ideally expressed only in terms of bounds on the partial correlations and the maximum degree. It is interesting to see that to large extent this is possible for attractive GGMs.

Weaknesses: Attractiveness is a very strong condition. To argue why the paper is a significant contribution to the literature, the authors should elaborate a bit more why this assumption is important from the practical point of view. Perhaps a good place to start is a recent paper: Agrawal, Raj, Uma Roy, and Caroline Uhler. "Covariance matrix estimation under total positivity for portfolio selection." arXiv preprint arXiv:1909.04222 (2019). Another weakness (but this is a growing problem overall) is that the whole interesting math is moved to the appendix and the paper became only a walk-through through the main ideas. I personally do not like this style as it is hard to get the feeling behind the structure of the problem without jumping back and forth between the paper and the appendix. Finally, below Lemma 1 you comment on other approaches to learning attractive GGMs. But you missed a paper in which actually some sample complexity results are provided: Yuhao Wang, Uma Roy, Caroline Uhler ; Proceedings of the Twenty Third International Conference on Artificial Intelligence and Statistics, PMLR 108:2698-2708, 2020. I think this has to be explicitly mentioned. This may involve some further changes in the paper. EDIT: I suggested motivating the paper by discussing where attractiveness appears because I felt this may be just very easy. But I actually agree with the fourth reviewer that it would be great to get a better feeling behind walk-summability. EDIT2: Thank you for clarification about the missing citation. Indeed, their result has a different flavour than yours. I felt this could be the only remaining issue but now I am happy to change my recommendation.

Correctness: I did not check the proofs carefully. I went through the first results in the appendix and it has a good flow so I generally believe in correctness of the paper.

Clarity: Yes, subject to some minor comments below.

Relation to Prior Work: Yes, I think the paper is really well linked to previous results (apart from my comment above).

Reproducibility: Yes

Additional Feedback: One page 4 you write that attractive GGMs are positively associated. I would try to be more precise because positive association has a meaning (Easy, Proschan,Walkup). You can write explicitly write that all partial correlations are nonnegative. On Page 4 the algorithm must be edited better. You must write that i is fixed. In Var(X_j|X_S,X_j) replace the first X_j with X_i. It is also not clear at what stage the value of t enters the picture. On page 6 there is "that that"


Review 2

Summary and Contributions: This paper gives the first fixed polynomial-time algorithms for learning attractive GGMs and walk-summable GGMs with a logarithmic number of samples when the precision matrix is ill-conditioned. The authors present the theoretical analysis of the proposed algorithms and empirically evaluate the effectiveness of the algorithms.

Strengths: The contribution is significant and novel: Under the assumption that the precision matrix is ill-conditioned, the recent work [14] provides an algorithm for learning GGMs with n^{O(d)} runtime which is inefficient. In this paper, the authors provide polynomial-time algorithms with theoretical guarantees for a class of GGMs under the same assumption. In addition, the proposed algorithms still have logarithmic sample complexity as [14]. They also prove information-theoretically optimal sample complexity for learning attractive GGMs, which previous literature leaves open. The experimental results show that the proposed methods can achieve much lower recovery errors than existing methods when the number of node is large given a fixed sample number.

Weaknesses: As this paper studies the runtime of the algorithms, the authors can also provide empirical comparison of the CPU time of different algorithms in the main text.

Correctness: The theoretical claims look correct. The detailed proofs are not carefully checked. The empirical methodology is correct.

Clarity: This paper is well written and well organized.

Relation to Prior Work: The authors have clearly compared their results with previous works.

Reproducibility: Yes

Additional Feedback:


Review 3

Summary and Contributions: The authors considered the problem of reconstructing the structure of gaussian graphical models (GGM) using an optimal number of samples. In particular they authors found subclasses of GGMs (walk-summable and positive GGMs) for which they designed low computational complexity algorithms that provably achieve optimal samples complexity.

Strengths: I believe this work is a very important theoretical contribution as GGM are the most fundamental probabilistic models and in my opinion these results are also quite relevant for practical applications (GGM are used everywhere in natural sciences where low sample complexity algorithm are often extremely important). I hope this paper will help the ML community to refocusing on finding optimal methods for learning arbitrary GGMs.

Weaknesses: This paper does not suffer from major weaknesses.

Correctness: The proofs look all correct to me.

Clarity: The paper is well-written, well-structured and easy to read.

Relation to Prior Work: I believe that the authors did an excellent job at putting their results into perspective and describing the previous contributions found in the litterature.

Reproducibility: Yes

Additional Feedback:


Review 4

Summary and Contributions: Setup. Fix kappa>0. Let Theta denote a square matrix such that - Theta is symmetric nonnegative positive definite - For every i,j with Theta_ij!=0, |Theta_ij|/sqrt(Theta_ii Theta_jj) > kappa. Given m i.i.d samples from N(0,Sigma), our task is to uncover the sparsity pattern of Theta. This paper develops a simple but effective algorithm to solve this problem, even if Theta is quite poorly conditioned.

Strengths: The method is simple, it appears to work, and it solves a problem that comes up all the time. The condition number problem is real: in practice two variables may be incredibly tightly correlated. It appears plausible that this approach works just as well in that case.

Weaknesses: Not all GPs are walk-summable. The end of the supplement gives some helpful intuition on what can go wrong. There's definitely still work to be done there to figure out how to know whether this algorithm is safe to apply...

Correctness: Looks plausible.

Clarity: Looks good. I would have included more simulations/experiments and buried more technical details in appendix. For example, all attractive gaussians are also walk-summable. So maybe you could just skip the in-depth discussion of the attractive gaussians. Mention them, but maybe not in so much detail. That would leave more room for showing some of the interesting examples where the method fails (because its not walk-summable). Dunno about other reviewers, but I always like to see at least some failure modes of a new method.

Relation to Prior Work: Seems clear.

Reproducibility: Yes

Additional Feedback: Minor -- the notation in line 136, I find "Var(Xj | Xs, Xj)" mysterious. In the supplement it is clear what you want to do, but the notation here makes it look like you want the variance of Xj given Xj... which is zero...

[Author Response · NeurIPS 2020]

**Author Feedback: Learning Some Popular Gaussian Graphical Models without Condition Number Bounds**

We thank the reviewers for their careful feedback. Below we answer the questions raised by the reviewers:

1. Reviewer 1 suggested adding some more citations to further motivate the study of attractive GGMs, which we plan to add, and asked how our results relate to the recent paper of Wang, Roy, and Uhler [3]. In the paper [3], the authors studied the learning of attractive GGMs when they are promised to be well-conditioned (Condition 3.1 in their paper), which they note is the same situation studied in the previous analysis of CLIME and other methods. Since the goal of our work is to study efficient learnability without a condition number assumption, the results in this work and their work (where they were interested in adaptivity, i.e. minimizing the number of tuning parameters) are orthogonal, and ours remains the first to give condition-number independent results for learning attractive GGMs with a computationally efficient algorithm.

2. Review 1 also asked that we elaborate on why attractiveness is important from a practical point of view. In fact [3] and earlier works studying the $MTP_2$ property address this point. In phylogenetic applications, observed variables are often positively dependent because of shared ancestry [4]. In various copula models that are popular in finance we posit a latent global market variable that also leads to positive dependence [2]. Finally the Gaussian free field, which plays a central role in some areas of mathematical physics, is a special case. We will add this discussion to our paper since it is clearly an important point of the broader context. In fact attractiveness in the context of Gaussian graphical models has been studied for almost forty years since the work of Karlin and Rinott [1].

3. A couple reviewers noticed a typo in the algorithm description on page 4: when we are learning the neighborhood of node $i$, step 2 of the GREEDYANDPRUNE algorithm looks for $j$ which minimizes $\widehat{\text{Var}}(X_i|X_S, X_j)$ (the typo is we replaced $i$ by $j$ here). As also noted by a reviewer, this is correct in the more detailed algorithm description given in the Appendix.

4. Reviewer 1 asked if the variable $t$ should enter into the loop in step 2 of GREEDYANDPRUNE. This is just a loop indexing variable because we want to run the greedy step $T$ times (so at each step, $S$ has size $t$).

5. There were some suggestions about what could be moved between the main body and the Supplementary Material, but the suggestions of different reviewers are currently in conflict (Reviewer 1 suggests more about attractive models and more technical content in the main body, Reviewer 4 suggests less of this and adding more of the experimental results). If the paper is accepted, hopefully there will be additional space to add some of this material.

6. Reviewer 2 asked about comparison of CPU runtimes. Though it is not provided in the main text of this version, it is in the supplementary (Table 2, which shows the sequential runtime is similar to other popular methods); we will add some mention of this to the main text. We note an advantage of this method over some alternatives like the graphical lasso is that it is "embarassingly parallelizable" (recovering the neighborhood of each vertex can be done in parallel).

7. The reviewers noticed a few other typos and suggested some other small edits for clarity — thanks, we plan to make these changes in the next version of this work.

# References

[1] Samuel Karlin, Yosef Rinott, et al. Total positivity properties of absolute value multinormal variables with applications to confidence interval estimates and related probabilistic inequalities. *The Annals of Statistics*, 9(5):1035–1049, 1981.

[2] Alfred Müller and Marco Scarsini. Archimedean copulae and positive dependence. *Journal of Multivariate Analysis*, 93(2):434–445, 2005.

[3] Yuhao Wang, Uma Roy, and Caroline Uhler. Learning high-dimensional gaussian graphical models under total positivity without adjustment of tuning parameters. In *International Conference on Artificial Intelligence and Statistics*, pages 2698–2708, 2020.

[4] Piotr Zwiernik. *Semialgebraic statistics and latent tree models*. CRC Press, 2015.


[Meta-Review · NeurIPS 2020]

This paper studies the well-known problem of structure learning Gaussian Graphical Models. This is simply the problem of learning the zero-non-zero structure of the "precision" matrix (inverse Covariance matrix) of an unknown gaussian distribution from samples. All known efficient algorithms for the problem suffer from a running time and sample complexity dependence on the condition number of the unknown covariance matrix. This paper gives an algorithm, which, for some covariance matrices that satisfy some structural assumptions (walk summabililty, attractive) gives an efficient algorithm that does not depend on the condition number of the unknown covariance (which can be arbitrarily ill-conditioned under the assumptions) . The reviewers (with clarifications in the author feedback) were convinced of both the motivation and non-triviality of the assumptions made and found the algorithmic contributions of this paper important. This paper makes progress on a fundamental problem in graphical model learning. I am pleased to recommend accepting it to the NeurIPS program.